# Preventing Latent Rehearsal Decay in Online Continual SSL with SOLAR

## Abstract

Continual learning methods enable models to learn from non-stationary data without forgetting. We study Online Continual Self-Supervised Learning (OCSSL), in which models learn from a continuous stream of unlabeled data. We find that OCSSL exhibits surprising learning dynamics, favoring plasticity over stability, with a simple FIFO buffer outperforming Reservoir sampling. We explain this result with the *Latent Rehearsal Decay* hypothesis, which attributes it to latent space degradation under excessive stability of replay. To quantify this effect, we introduce two metrics (*Overlap* and *Deviation*) and show their correlation with declines in probing accuracy. Building on these insights, we propose *SOLAR*, which leverages efficient online proxies of *Deviation* to guide buffer management and incorporates an explicit *Overlap* loss. Experiments demonstrate that *SOLAR* achieves state-of-the-art performance on OCSSL vision benchmarks, highlighting its effectiveness in balancing convergence speed and final performance.

## 1 Introduction

Continual learning (CL) addresses the fundamental challenge of enabling machine learning models to acquire new knowledge from sequential tasks while preserving previously learned knowledge (Parisi et al., 2019; De Lange et al., 2022; McCloskey & Cohen, 1989; Kirkpatrick et al., 2017). In online CL (Soutif-Cormerais et al., 2023; Mai et al., 2022; Parisi & Lomonaco, 2020; Lopez-Paz & Ranzato, 2017), data is seen as a continuous one-pass stream of small minibatches, precluding multi-epoch training. This constraint favors rapid adaptation (Caccia et al., 2022; Hammoud et al., 2023). Online CL methods use Replay, paired with buffers that provide an unbiased sample of the stream, most notably the Reservoir buffer (Vitter, 1985). Concurrently, Self-Supervised Learning (SSL) has emerged as a powerful paradigm for representation learning that does not rely on labeled data (Chen et al., 2020; He et al., 2020; Grill et al., 2020; Zbontar et al., 2021). In this paper, we focus on *Online Continual Self-Supervised Learning* (OCSSL) (Yu et al., 2023; Cignoni et al., 2025a;b), a challenging learning scenario that combines the temporal constraints of online learning with the label-free nature of SSL. This scenario reflects many real-world applications in which unlabeled data streams continuously arrive and storage constraints prevent retention of historical data. It is especially relevant in online CL, where it is unrealistic to assume that labels—particularly human-provided ones—will be available for a continuous data stream. Consequently, the ability to learn high-quality unsupervised representations becomes essential.

Traditionally, CL has mitigated forgetting with regularization losses (Urettini & Carta, 2025; Zenke et al., 2017) that explicitly constrain model updates relative to past states, thereby preserving stability (retention of past knowledge) at the cost of reduced plasticity (adaptation to new data). These approaches are often coupled with replay buffers; the buffer policy itself further modulates this stability-plasticity trade-off: Reservoir (Vitter, 1985) emphasizes stability by maintaining long-term memory, whereas FIFO (Isele & Cosgun, 2018) favors plasticity by discarding older samples.

In this paper, we show that OCSSL methods struggle with plasticity and long-term learning, shifting the primary challenge from stability to optimal plasticity. Intuitively, a CL method would be expected to forget more with longer training, favoring stability-focused solutions. Surprisingly, we find that, in scenarios needing plasticity, stability-focused solutions (Reservoir) fail on long training schedules compared to a naive FIFO-based solution. We demonstrate that this unexpected result is caused by a novel collapse phenomenon: **Latent Rehearsal Decay**, which arises with prolonged

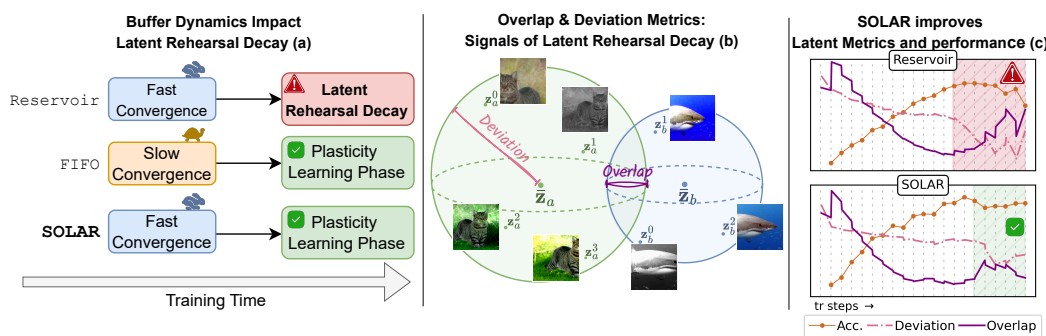

Figure 1: **Motivation and overview.** (**a**) Reservoir buffer, traditionally considered stability-focused, converges faster than FIFO but leads to degraded latent structure, manifesting as a phenomenon we call *Latent Rehearsal Decay*. (**b**) SSL methods can be viewed as solving an instance-discrimination problem: each sample and its augmentations form a *hyperball* in the embedding space. The spread of each hyperball (*Deviation*) and their pairwise *Overlap* are metrics that reveal the deterioration in feature space quality during training caused by Latent Rehearsal Decay. (**c**) Our proposed *SOLAR* method improves both Deviation and Overlap latent metrics, yielding better overall performance.

training on a static subset of data, as occurs at the limit with Reservoir buffers (Figure 1(a)). This leads to a degraded, overspecialized latent space that hinders adaptation to new tasks. This latent space degradation manifests as performance drops in longer training schedules and can be detected by two novel latent metrics *Deviation* and *Overlap* (Figure 1(b)).

We introduce **SOLAR (Self-supervised Online Latent-Aware Replay)**, a novel strategy that applies implicit and dynamic regularization by enforcing the quality of the latent space without explicitly constraining network updates. *SOLAR* combines a *Deviation-Aware Buffer* and an *Overlap Loss* that prevent latent rehearsal decay by implicitly optimizing Deviation and Overlap via efficient online proxies (Figure 1(c)). Through extensive experiments we demonstrate that our approach successfully adapts to unknown training lengths while avoiding Latent Rehearsal Decay.

## 2 ONLINE CONTINUAL SELF-SUPERVISED LEARNING (OCSSL)

In SSL, an encoder network $f : \mathcal{X} \to \mathcal{F}$ is trained to map an input $x \in \mathcal{X}$ to a feature representation $z \in \mathcal{F}$ by solving pretext tasks requiring no labels (Ericsson et al., 2022). We employ the popular class of SSL methods using *instance discrimination* pretext tasks (Gui et al., 2024). Two different augmented views, $x_1$ and $x_2$, are generated from the same sample, then the views are passed through $f$ and usually through a projector network (Chen & He, 2021) that maps encoded views into a projection space $\mathcal{P}$. The pretext task consists in enforcing the two projected views in $\mathcal{P}$ to be close in the feature space. Following other works in continual SSL (Purushwalkam et al., 2022; Cignoni et al., 2025b; Li et al., 2022), we employ SimSiam (Chen & He, 2021) as the SSL method of choice.

OCSSL is a stream learning paradigm in which unlabeled data arrive sequentially, sharing traits with online (supervised) CL (Soutif-Cormerais et al., 2023; Mai et al., 2022), but with challenges unique to SSL. The data stream $\mathcal{D}$ induces a dual-level structure in the learning process. At a broader scale, $\mathcal{D}$ is divided into class-incremental distributions (i.e. tasks) $\mathcal{D}_T$ which are unknown to the model, resulting in a *task-agnostic* scenario. Instead, at the granular level these tasks present themselves as streamed minibatches. The model receives a minibatch $x_t^T \in \mathcal{D}_T$ at each timestep $t$. Each $x_t^T$, characterized by a fixed, small batch size $b_s$ (usually ranging from 1 to 10), and becomes permanently inaccessible once processing advances to the next minibatch. Although multiple training passes for each streamed minibatch are feasible, this online scenario naturally precludes multi-epoch training. Performance is commonly evaluated by training a linear classifier on frozen representations (linear probing) (Alain & Bengio, 2016), as adopted in others OCSSL methods (Yu et al., 2023; Cignoni et al., 2025b; Purushwalkam et al., 2022); we follow the same protocol.

**Related Work.** Replay methods are the dominant paradigm in Online CL (Soutif-Cormerais et al., 2023), since small streaming minibatches are insufficient for effective learning – especially in SSL, for which large batches are crucial (Chen et al., 2020; Zbontar et al., 2021). Replay buffers alleviate

this by providing additional samples. Reservoir buffer is the de facto standard (Mai et al., 2022; Buzzega et al., 2020; Wang et al., 2024; Rolnick et al., 2019), as it maintains an unbiased subset of the stream. FIFO buffers are also used, although less commonly (Cignoni et al., 2025b; Cai et al., 2021).

Most continual self-supervised learning methods are designed for the *offline* setting in which *multi-epoch* training on each experience is possible and *task boundaries* are known. They typically rely on distillation to mitigate forgetting and contrastive losses on multiple views, such as CaSSLe (Fini et al., 2022) and PFR (Gomez-Villa et al., 2022), and extensions like SyCON (Cha & Moon, 2023), Osiris (Zhang et al., 2025) and POCON (Gomez-Villa et al., 2024).

Only a few works explicitly address the OCSSL scenario, which is *task-agnostic* and restricted to *single-epoch* training on the stream. SCALE extends SimCLR using an InfoNCE-like loss (Oord et al., 2018). It uses distillation between old and current feature representations and updates its buffer using the Part and Select Algorithm (PSA) (Yu et al., 2023). CMP proposes a replay-free approach, augmenting mini-batches with multiple patches (Cignoni et al., 2025a), while MinRed performs exemplar replay with maximally correlated samples (Purushwalkam et al., 2022). CLA employs distillation through a temporal projector, using either an EMA teacher (CLA-E) or stored past latent features (CLA-R), and introduces plasticity with a FIFO buffer (Cignoni et al., 2025b).

State-of-the-art OCSSL relies on replay buffers. In the next section, we analyze the behavior of the widely used Reservoir and the less common FIFO buffer over long OCSSL training sessions.

## 3 RESERVOIR AND FIFO: STABILITY–PLASTICITY IN LONG TRAINING

In this section we compare Reservoir and FIFO buffers. Reservoir encourages *stability* by providing an unbiased set of samples, while FIFO encourages *plasticity* (Kobayashi, 2025; Isele & Cosgun, 2018). We show that FIFO outperforms Reservoir in long training schedules, which suggests a currently unexplored failure mode of replay methods.

**Reservoir versus FIFO Buffers.** Reservoir sampling ensures that each incoming sample has equal probability of being included in the buffer $\mathcal{M}$. Once the buffer is full, the $t$-th incoming sample $x_t$ is inserted with probability $\frac{|\mathcal{M}|}{t}$: a random index $i = \text{rand}(0, t)$ is drawn, and if $i \leq |\mathcal{M}|$ the entry $\mathcal{M}[i]$ is replaced by $x_t$; otherwise $x_t$ is discarded. Reservoir buffers encourage *stability* by providing an unbiased view of the stream. Notice that the insertion probability $\frac{|\mathcal{M}|}{t}$ decays over time. As a result, it will converge to a static subset of the stream and revisit old samples more often. In contrast, FIFO retains only the $|\mathcal{M}|$ most recent samples, discarding the oldest as new ones arrive. This encourages *plasticity*, since each training step uses only the latest samples. Consequently, each sample is stored for the same duration and revisited equally often.

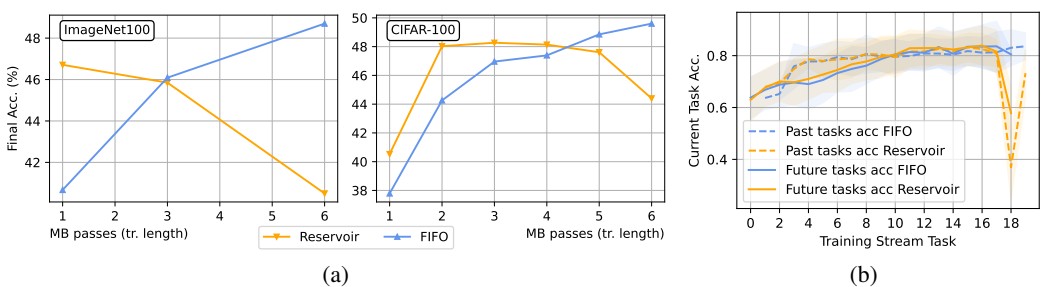

(a)                                    (b)

Figure 2: Impact of training length. (a) Reports the Final Accuracy for Reservoir and FIFO, on ImageNet-100 and CIFAR-100, trained on an OCSSL stream with varying training length. Reservoir converges faster for shorter schedules, but suffers unusual performance drops for longer schedules. Training length is expressed as number of minibatch passes (i.e. training steps) per incoming stream minibatch. (b) Gives the average per-task probing accuracy on past and future tasks relative to the current task (CIFAR-100). Past task accuracy does not decrease and both FIFO and Reservoir achieve similar results, excluding final Reservoir drop (explained later), implying that forgetting has little impact in OCSSL.

**Failure of Reservoir under Long Training Schedules.** Given this view, one might expect Reservoir to perform better under longer training, where stability helps avoid forgetting. However, Figure 2(a) shows the opposite trend. As training length increases, Reservoir accuracy drops, while FIFO steadily improves and surpasses it. A closer look at past and future performance (Figure 2(b)) reveals why. In the final phases of the training, Reservoir not only forgets previous tasks, but also generalizes poorly to new ones. In contrast, FIFO maintains both stability and plasticity, generalizing well to both past and future tasks. More details and per-task accuracies are provided in Appendix A. The failure of Reservoir under long training schedules suggests some latent causes unrelated to forgetting. In the next section, we show that this unexpected behavior arises from a phenomenon we call *Latent Rehearsal Decay*.

## 4 LATENT REHEARSAL DECAY

*Can we explain the underperformance of Reservoir buffers over long training schedules?*

**Hypothesis** (Latent Rehearsal Decay). *We hypothesize that long training schedules on a limited subset of data lead to overfitting, which degrades the feature space by producing overspecialized representations. These representations hinder adaptability to new tasks, resulting in degradation of probing accuracy.*

This hypothesis suggests that, in certain scenarios where plasticity is critical, reservoir sampling convergence leads to suboptimal learning dynamics. Introducing greater diversity in later training stages can prevent convergence to suboptimal minima. FIFO achieves this via higher plasticity (Figure 2b), though at the cost of slower convergence. Notice that prior work links collapse in SSL to feature collapse (Li et al., 2022) or lack of uniformity in the latent space (Wang & Isola, 2020). We show in Appendix B.3 that these phenomena are unrelated to Latent Rehearsal Decay in OCSSL and in Appendix B.4 that it differs from simple buffer overfitting.

To verify the hypothesis, we introduce two novel metrics inspired by intra- and inter-class metrics in the supervised setting (Sui et al., 2025). Our metrics assess feature quality via *inter-sample* relationships across different samples and *intra-sample* relationships across views of the same sample. Let $\mathbb{A}$ be the set of augmentations; then, for each sample $x_a$ in the training stream, we define its set of all possible augmented feature views in the latent space as $\mathbb{T}_a = \{\mathbf{z}_a^i | \mathbf{z}_a^i = f(x_a^i)\}_{i \in \mathbb{A}}$, where each $x_a^i$ is a different augmented view of the original $x_a$ and $f$ is the encoder network. Then, each $\mathbb{T}_a$ identifies a *hyperball* in the feature space centered on the mean feature view $\bar{\mathbf{z}}_a = \frac{1}{|\mathbb{T}_a|} \sum_{\mathbf{z}_a^i \in \mathbb{T}_a} \mathbf{z}_a^i$ and encompassing feature views (Figure 1(b)). We now propose two new metrics, the *Deviation* and the *Overlap* which characterize Latent Rehearsal Decay.

**Deviation.** This intra-sample metric measures the distance in feature space between multiple views of a single sample $x_a$ by calculating the average pairwise cosine distance of feature views $\mathbf{z}_a^i$ in $\mathcal{P}$. Let $S_C$ be the cosine similarity function. We define the deviation as:

$$\text{Dev}(\mathbb{T}_a) = \frac{1}{|\mathbb{T}_a|^2} \sum_{\mathbf{z}_a^i, \mathbf{z}_a^j \in \mathbb{T}_a} \left(1 - S_C(\mathbf{z}_a^i, \mathbf{z}_a^j)\right). \tag{1}$$

**Overlap.** This inter-sample metric measures the overlap in the latent space between feature views of different samples. More precisely, for each hyperball $\mathbb{T}_a$ we consider its average feature $\bar{\mathbf{z}}_a$ and the average angle of the pairwise cosine similarity ($S_C$) of all feature views:

$$\bar{\theta}_a = \frac{1}{|\mathbb{T}_a|^2} \sum_{\mathbf{z}_a^i, \mathbf{z}_a^j \in \mathbb{T}_a} \arccos\left(S_C(\mathbf{z}_a^i, \mathbf{z}_a^j)\right), \tag{2}$$

calculated in the encoder space $\mathcal{F}$. We define the Overlap between two samples $x_a$ and $x_b$ as:

$$Ov(\mathbb{T}_a, \mathbb{T}_b) = \left(\bar{\theta}_a + \bar{\theta}_b\right) - \theta\left(\bar{\mathbf{z}}_a, \bar{\mathbf{z}}_b\right), \tag{3}$$

where $\theta\left(\bar{\mathbf{z}}_a, \bar{\mathbf{z}}_b\right)$ denotes the angle between average features $\bar{\mathbf{z}}_a$ and $\bar{\mathbf{z}}_b$. The Overlap effectively measures the distance between the two hyperballs. When $Ov(\mathbb{T}_a, \mathbb{T}_b) > 0$ we effectively have an intersection between the two samples. Since we want to count only intersecting hyperballs, given a

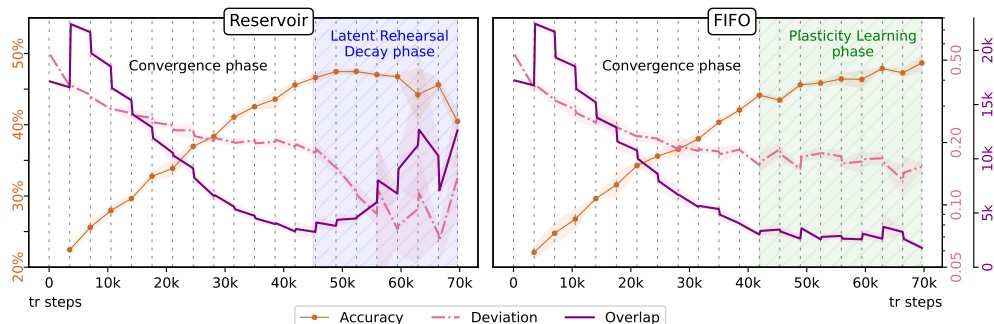

Figure 3: **Latent Rehearsal Decay**. Accuracy and metrics calculated on the entire training stream of ImageNet100. Both FIFO and Reservoir have an initial *Convergence phase*, where Accuracy increases and both metrics improve. After a certain point, Reservoir transitions into the *Latent Rehearsal Decay phase*, where the sudden drop in accuracy is preceded by a sharp decrease of Deviation and increase of Overlap. Further analysis on CIFAR-100 in Appendix B.2.

dataset $D = \{\mathbb{T}_1, \ldots, \mathbb{T}_{|D|}\}$, the *Average Overlap Count* for the set is calculated by:

$$\overline{\text{Ov}}_{\text{count}}(D) = \frac{1}{|D|^2} \sum_{\mathbb{T}_i, \mathbb{T}_j \in D} \mathbf{1}\left[Ov(\mathbb{T}_i, \mathbb{T}_j) > 0\right]. \tag{4}$$

Our metrics, differing from existing ones, do not only have a global view of the feature space, but also consider the local latent behavior relative to each sample, which is essential to capture a more fine-grained view of representation quality and convergence of SSL methods. We do this by considering the hyperballs defined by feature space views, which give a local insight even for an inter-sample metric such as Overlap.

**Metric Analysis and Comparisons.** Figure 3 shows the probing accuracy, Deviation, and Overlap metrics during training for Reservoir and FIFO. Specifically, given the entire training stream $\mathcal{D}$, we extracted $\overline{\text{Ov}}_{\text{count}}(\mathcal{D})$ (Eq. (4)), and $\frac{1}{|\mathcal{D}|} \sum_{\mathbb{T}_i \in \mathcal{D}} \text{Dev}(\mathbb{T}_i)$. Metrics are calculated offline by sampling 20 augmentations from each $\mathbb{T}_a$, for all samples in the entire training stream $\mathcal{D}$. During the initial *Convergence phase*, both methods continuously improve as evidenced by the growing accuracy, decreasing Overlap, and slight decrease in Deviation. Subsequently, the behavior of the two methods diverges. Reservoir suffers from *Latent Rehearsal Decay*, which we observe as a sudden drop of Deviation and increase in Overlap. The degradation of the latent space is followed by a drop in the probing accuracy. Instead, FIFO continuously improve (*Plasticity Learning phase*) in probing accuracy and does not exhibit degradation of the latent space. This suggests that the reason behind the decay lies in the Reservoir strategy, where in the later training stages the buffer consists of a slowly changing set of low-deviation samples. Further training on this set leads to a degraded latent space and lower generalization. Relation between training loss and metrics in Appendix B.1.

## 5 Self-supervised Online Latent-Aware Replay (SOLAR)

Our analysis reveals that an optimal OCSSL solution should be both plastic and stable, adapting the tradeoff dynamically depending on training length. Our goal, therefore, is to develop an adaptive mechanism that performs well at any point during the training process, *regardless* of the training length, converging fast for short training, while rivaling FIFO plasticity in longer schedules.

Our method Self-supervised Online Latent-Aware Replay (SOLAR) achieves this goal by preventing *Latent Rehearsal Decay*. This is done via efficient online proxies for the *Deviation* and *Overlap*. Specifically, SOLAR has 2 components: (1) the **Deviation-Aware Buffer**, that controls Deviation by prioritizing high-Deviation samples in the buffer; (2) the **Overlap Loss**, that penalizes overlap between new and buffer samples. The full algorithm is given in Appendix C.1.

### 5.1 Deviation-Aware Buffer

A central objective of SOLAR is to prevent the Deviation collapse, which characterizes Latent Rehearsal Decay (see Figure 3, left). SOLAR addresses this by employing a memory buffer that replays

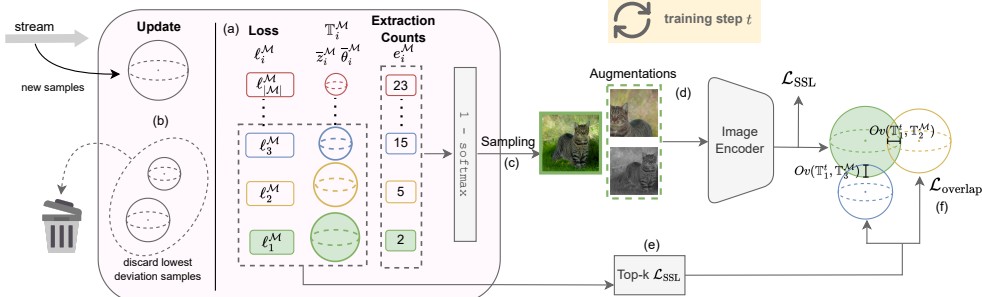

Figure 4: **Overview of SOLAR.** (**a**) The deviation-aware buffer tracks average representation, loss, and extraction count for each sample; (**b**) Low-deviation representations are discarded; (**c**) Sampling is inversely proportional to extraction counts; (**d**) Sampled images are augmented and used for SSL training with the loss $\mathcal{L}_{SSL}$; (**e**) The top-$K$ samples with highest loss are extracted to compute the Overlap loss $\mathcal{L}_{\text{overlap}}$ (**f**).

high-Deviation samples, thereby increasing the diversity of feature representations during training and avoiding the sudden drop in Deviation. At first glance, this requires online monitoring of sample Deviation. However, as shown in Appendix C.2 the self-supervised loss itself is a good proxy since it is positively related to Deviation:

$$\frac{d\,\text{Dev}}{d\mathcal{L}_{SSL}} = \frac{1}{n^2} > 0\,, \tag{5}$$

where $n$ is the number of augmented views. Hence, samples with higher SSL loss (i.e. less converged) correspond to higher-Deviation examples, whereas low-loss samples (i.e., already learned) correspond to lower Deviation ones.

The memory buffer $\mathcal{M}$ is a list in which each entry stores a sample as a tuple $\langle x_i^{\mathcal{M}}, \ell_i^{\mathcal{M}}, \bar{\mathbf{z}}_i^{\mathcal{M}}, \bar{\theta}_i^{\mathcal{M}}, e_i^{\mathcal{M}} \rangle$ (Figure 4(a)), where $x_i^{\mathcal{M}}$ is the input, $\ell_i^{\mathcal{M}}$ an estimate of the loss computed in previous iterations, $\bar{\mathbf{z}}_i^{\mathcal{M}}$ its average representation, $\bar{\theta}_i^{\mathcal{M}}$ its average angle, and $e_i^{\mathcal{M}}$ a counter for the number of times a buffer sample is extracted for training.

At time $t$, SOLAR trains on a batch of samples coming from the buffer $\{x_i^t\}_{i=1...B_{tot}}$ and computes their self-supervised loss $\ell_i^t$, average feature representation $\bar{z}_i^t$ across augmentations, and the corresponding average angle of the pairwise cosine similarities among augmented features views $\bar{\theta}_i^t$ (as defined in Eq. (2)). If unseen stream samples are available, they are used for training and then inserted into the buffer. Once the maximum capacity is reached, the samples discarded are the ones with minimal loss $\ell_i^t$ (Figure 4(b)), a criterion that encourages replay of high-Deviation samples (see Appendix C.2). When a memory sample $i$ is extracted from the buffer, its statistics are updated using an Exponential Moving Average (EMA) with decay factor $\eta = 0.5$:

$$\ell_i^{\mathcal{M}} \leftarrow \eta \cdot \ell_i^{\mathcal{M}} + (1-\eta) \cdot \ell_i^t, \quad \bar{z}_i^{\mathcal{M}} \leftarrow \eta \cdot \bar{z}_i^{\mathcal{M}} + (1-\eta)\bar{z}_i^t, \quad \bar{\theta}_i^{\mathcal{M}} \leftarrow \eta \cdot \bar{\theta}_i^{\mathcal{M}} + (1-\eta)\bar{\theta}_i^t. \tag{6}$$

This approach of EMA-updated buffer statistics has been already employed in OCSSL by Purush-walkam et al. (2022); Cignoni et al. (2025b), the motivation being to make the statistics more robust to sudden distribution shifts of CL which could impair their reliability. Its effect is ablated in Appendix C.5.

At each iteration, samples in $\mathcal{M}$ are replayed according to a *deviation-aware extraction policy* (Figure 4(c)). Let $\bar{e}_i^{\mathcal{M}} \in [0,1]$ denote the normalized extraction counts across all buffer samples; the sampling probabilities are:

$$\mathbf{p}_{[1,...,|\mathcal{M}|]} = 1 - \text{Softmax}(\bar{\mathbf{e}}_{[1,...,|\mathcal{M}|]}).$$

Thus, samples with lower counters (i.e., replayed fewer times) are more likely to be selected. Intuitively, these samples are under-trained and thus have higher Deviation. Emphasizing them during training enhances sample diversity and introduces a balancing effect that prevents bias toward a small subset of well-learned samples. We regard this mechanism as complementary to the removal of minimal-loss samples; using them together promotes training on high-Deviation data. Different extraction policies are ablated in Appendix C.5. Once a sample is selected, it is augmented into

multiple views and passed through the backbone (Figure 4(d)) to compute the SSL loss ($\mathcal{L}_{SSL}$) and the SGD step, and then its counter is incremented by one.

In-depth comparison with other methods prioritizing "hard" samples is presented in Appendix C.6.

## 5.2 OVERLAP LOSS

Solely prioritizing higher-deviation samples during training is not sufficient to mitigate Latent Rehearsal Decay, as another contributing factor is Overlap, which increases as the phenomenon occurs (see Figure 3, left). In order to approximate overlap online, we leverage the average features $\bar{\mathbf{z}}^t$ and cosine angles $\bar{\theta}^t$ between two augmentations of the same sample, which are readily available during SSL training on the stream at timestep $t$ and stored in the buffer as discussed above (further analysis on online overlap approximation in Appendix C.3).

During training, we minimize the overlap between the current minibatch and the top-$K$ highest-deviation buffer samples not used for the current minibatch. The rationale is that the highest deviation samples are those occupying more space in the latent space, and thus more probable to have overlap with other, especially newer, samples. Once again, we employ the per-sample loss to estimate Deviation and select the top-$K$ highest-loss samples from the buffer (Figure 4(e)). The Overlap loss is:

$$\mathcal{L}_{\text{overlap}} = \frac{1}{b} \sum_{i=1}^{b} \frac{1}{K} \sum_{k=1}^{K} \max\left(0, Ov(\mathbb{T}_i, \mathbb{T}_k^{\mathcal{M}})\right), \tag{7}$$

where $b$ is the mini-batch size and $Ov(\mathbb{T}_i, \mathbb{T}_k^*)$ is defined as in Eq. (3) (Figure 4(f)). Each $\mathbb{T}_k^{\mathcal{M}}$ is composed by *frozen* $\bar{\mathbf{z}}_k^{\mathcal{M}}$ and $\bar{\theta}_k^{\mathcal{M}}$ extracted from $\mathcal{M}$, while $\mathbb{T}_i$ features $\bar{\mathbf{z}}_i$ and $\bar{\theta}_i$ are computed online from the two views for each sample in the minibatch and thus contribute to training the backbone. The $\max$ operator ensures that only positive overlaps are penalized. In our experiments we set $K = 500$. In a sense, $\mathcal{L}_{\text{overlap}}$ is like a *targeted contrastive loss* that pushes away samples only as much as is needed and contrasting only with the most problematic samples. This overlap loss encourages the model to differentiate features that are overly aligned, thus preserving representational diversity across the buffer and training stream. The overall training objective of SOLAR is $\mathcal{L}_{SSL} + \omega \mathcal{L}_{\text{overlap}}$, where $\omega$ is a hyperparameter. Analysis on $\omega$ is provided in Appendix C.4).

## 6 EXPERIMENTS

In this section we compare SOLAR with the state-of-the-art and ablate and analyze its components.

**Experimental Setup.** We conducted experiments on the Split CIFAR-100 (Krizhevsky et al., 2009) and Split ImageNet100 (Deng et al., 2009) class-incremental benchmarks with 20 experiences each, and also on CLEAR100 (Lin et al., 2021), a domain incremental learning benchmark. All benchmarks were presented to the models as an OCSSL stream with a stream minibatch of 10. All methods were allowed to extend the minibatch to size 138 from the buffer to maintain a fair comparison. The maximum buffer size $|\mathcal{M}|$ was set to 2000. Following other continual SSL works (Purushwalkam et al., 2022; Cignoni et al., 2025b;a), we use SimSiam (Chen & He, 2021) as the base SSL method on ResNet-18 (He et al., 2016). In the main experiments (Table 1), we simulated a training schedule with 6 minibatch passes for each incoming stream minibatch. More training protocol details are in Appendix D.1. Code to reproduce the results will be released upon acceptance.

**Metrics.** Evaluation was conducted by linear probing on the entire dataset. Probing is performed offline, as the goal of OCSSL is *representation learning*, and linear probing on a classification task is the standard measurement of this latent space quality in SSL (Ericsson et al., 2022). We report two probing metrics: FINAL ACCURACY, which is the probing accuracy at the end of the stream; and AVERAGE ACCURACY, which is the average of probing accuracies calculated at the end of each experience. Average Accuracy is a coarse measure of how good the model is throughout the training process (a proxy for Average Anytime Accuracy (Soutif-Cormerais et al., 2023)) and, for this reason, a good indication of fast convergence. Naturally, a model that is able to converge faster and retain learned knowledge will have higher Average Accuracy than a model which is slower to converge, even if they have comparable Final Accuracy. On the other hand, Final Accuracy is a good metric for the model performance on long training schedules, as it is sensitive to the sudden drops of accuracy typical of Latent Rehearsal Decay.

Table 1: Results on streaming online CIFAR-100 (20 experiences), ImageNet-100 (20 experiences), and CLEAR100 (11 experiences). Best in **bold**, second best underlined. We report mean and std over 3 runs.

| METHOD | BUFFER | DISTILL. | CIFAR-100 (20 exps) | | ImageNet100 (20 exps) | | CLEAR100 (11 exps) | |
|---|---|---|---|---|---|---|---|---|
| | | | FINAL ACC. | AVG. ACC. | FINAL ACC. | AVG. ACC. | FINAL ACC. | AVG. ACC. |
| ER | Reservoir | ✗ | $44.4 \pm 1.0$ | $39.6 \pm 0.3$ | $40.5 \pm 1.5$ | $39.3 \pm 0.4$ | $47.1 \pm 0.2$ | $35.3 \pm 0.4$ |
| ER | FIFO | ✗ | $49.0 \pm 0.3$ | $39.3 \pm 0.2$ | $48.7 \pm 1.0$ | $38.9 \pm 0.3$ | $45.3 \pm 1.8$ | $34.9 \pm 0.5$ |
| MinRed | MinRed | ✗ | $46.5 \pm 0.3$ | $40.9 \pm 0.3$ | $48.0 \pm 0.3$ | $42.5 \pm 0.2$ | $51.3 \pm 0.1$ | $39.7 \pm 0.1$ |
| LARS | LARS | ✗ | $47.0 \pm 0.7$ | $39.5 \pm 0.2$ | $43.2 \pm 2.8$ | $39.2 \pm 0.2$ | $44.7 \pm 0.9$ | $34.1 \pm 0.1$ |
| PER | PER | ✗ | $48.5 \pm 0.2$ | $39.1 \pm 0.2$ | $48.8 \pm 0.6$ | $38.8 \pm 0.2$ | $46.0 \pm 0.5$ | $35.0 \pm 0.2$ |
| SCALE | PSA | ✓ | $31.9 \pm 0.3$ | $27.1 \pm 0.3$ | $36.7 \pm 0.2$ | $29.8 \pm 0.1$ | $44.2 \pm 0.3$ | $41.0 \pm 0.3$ |
| CLA-E | FIFO | ✓ | $45.6 \pm 0.4$ | $34.2 \pm 0.2$ | $49.0 \pm 0.5$ | $36.6 \pm 0.2$ | $37.7 \pm 1.8$ | $29.3 \pm 1.4$ |
| CLA-R | FIFO | ✓ | $46.7 \pm 0.5$ | $42.3 \pm 0.3$ | $43.1 \pm 4.6$ | $42.0 \pm 0.2$ | $46.7 \pm 0.4$ | $35.3 \pm 0.2$ |
| SOLAR | Deviation-Aware | ✗ | **$49.5 \pm 0.5$** | **$42.3 \pm 0.3$** | **$49.4 \pm 1.5$** | **$42.8 \pm 0.2$** | **$51.5 \pm 0.8$** | **$41.3 \pm 0.4$** |

## 6.1 COMPARISON WITH THE STATE-OF-THE-ART

We see in Table 1 that SOLAR achieves state-of-the-art performance on all three benchmarks. Methods that perform explicit distillation (CLA and SCALE) fail to achieve consistently good results. SCALE, which employs both distillation and a stability-biased buffer, severely underperforms on both ImageNet100 and CIFAR-100, underscoring the need for plasticity in OCSSL. CLA-E and CLA-R highlight the downsides of fixed explicit distillation, despite both relying on a FIFO buffer.

CLA-E achieves good Final Accuracy on CIFAR and ImageNet, but converges very slowly with lower Average Accuracy and a flatter accuracy training curve (Appdx. D.2), similar to FIFO. This suggests that CLA-E regularization is too biased towards plasticity at the expense of fast convergence. In contrast, CLA-R converges faster and achieves Average Accuracy closer to SOLAR, but suffers from Latent Rehearsal Decay in later phases (see Appendix D.2). This is significant because even though CLA-R uses a plasticity-focused FIFO buffer it behaves as an overly stable strategy, demonstrating that Latent Rehearsal Decay can also be induced by excessive distillation. It seems that fixed distillation, like FIFO and Reservoir buffers, is not ideal for OCSSL as it is not capable of adapting to changing requirements during training.

MinRed is the only other method that maintain relatively high Final and Average Accuracy across all benchmarks. MinRed shows decently fast convergence. Like SOLAR, it leverages latent space information to manage its buffer, again demonstrating the suitability of this approach for OCSSL.

CLEAR100 exhibits dynamics quite different from CIFAR-100 and ImageNet100, favoring stability-biased methods because it is not a class-incremental stream. We hypothesize that, in a domain-incremental scenario with comparatively weaker shift such as CLEAR, the need for plasticity to adapt to novel data is reduced. For example, FIFO unexpectedly performs worse than Reservoir in both Average and Final Accuracy, while CLA-E performs poorly,

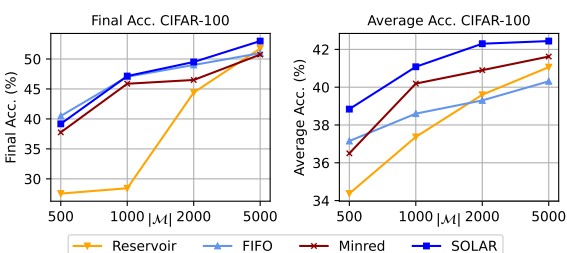

Figure 5: Impact of buffer size.

likely due to underfitting. On the other hand, SCALE is more competitive on CLEAR100 – especially in Average Accuracy – as it is stability-focused. Nonetheless, SOLAR is able to perform well in this scenario thanks to its adaptive plasticity capabilities.

Across all three benchmarks, PER and LARS exhibit performance comparable to FIFO and Reservoir, respectively. For PER, this indicates that prioritizing samples only at extraction time has limited impact when the underlying buffer composition remains unchanged. For LARS, although deletion is guided by a loss-based criterion, new samples are still inserted according to the standard Reservoir rule. Consequently, the insertion probability decays over time, causing the buffer to converge to a fixed subset – mirroring the behavior of Reservoir and leading to similar performance.

Overall, the advantage of SOLAR over the other methods is simultaneously maintaining high Average and Final Accuracy. While other methods can achieve comparable results in *one* of the metrics, they either fail to converge fast or suffer from Latent Rehearsal Decay.

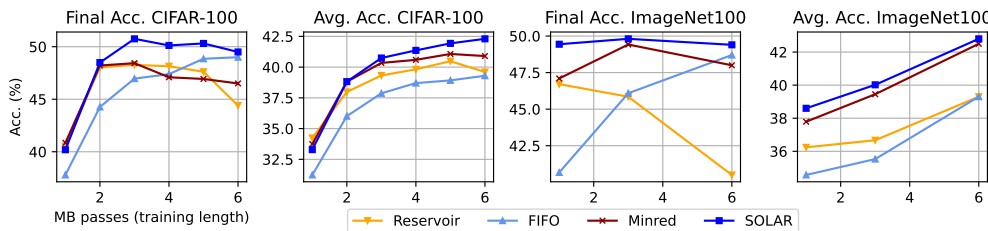

Figure 6: Performance at different training lengths. SOLAR outperforms competitors in both metrics independently of training length.

## 6.2 ABLATIONS AND ANALYSIS

**Changing Training Length.** Figure 6 reports plots for Final and Average Accuracy when training with shorter schedules (i.e. reducing the number of minibatch passes). Again, SOLAR outperforms the state-of-the-art for most training lengths, except for the shortest on CIFAR; here SOLAR is almost on par with methods like Reservoir and MindRed that focus on stability (and thus converge faster). Nonetheless, we note both those methods decrease in Final Accuracy when training is longer, likely due to Latent Rehearsal Decay. SOLAR shows clear advantages on ImageNet100, particularly in Final Accuracy where performance remains stable across training lengths and thus demonstrating its effectiveness as a length-agnostic strategy.

**Changing Buffer Size.** Figure 5 gives plots for Final and Average Accuracy when training with different buffer sizes on CIFAR-100. We see that Reservoir is penalized by reduced buffer dimensions, obtaining particularly low Final Accuracy for the two smallest sizes. This low performance is evidence of Latent Rehearsal Decay, with the buffer size directly exacerbating this phenomenon for Reservoir, as shown by the training curves in Figure 7. In fact, the drop in accuracy is inversely proportional to the buffer size, which is coherent with our hypothesis of Latent Rehearsal Decay being directly linked to subset overfitting for Reservoir. As expected, FIFO is less affected by decreased buffer size, as it has a strong bias towards recent samples. MinRed maintains good performance in Final Accuracy, but falls short in Average Accuracy when the buffer size is small ($|\mathcal{M}| = 500$), indicating that it is not able to maintain fast convergence under such constraints. SOLAR achieves rapid convergence across buffer sizes, surpassing other methods by far in Average Accuracy, while keeping Final Accuracy close to FIFO. The advantage of SOLAR in being able to maintain fast convergence independently from the buffer size makes it ideal even for OCSSL scenarios with generous buffer dimension. We observe similar results on ImageNet (see Appendix D.3).

**Ablations.** In Table 3 we ablate the SOLAR components. The Deviation buffer improves convergence compared to FIFO and Reservoir buffers, as reflected in the higher Average Accuracy. The Overlap loss, however, does not significantly improve convergence – Average Accuracy is comparable with or without it – but it substantially enhances Final Accuracy. This indicates that the Overlap loss plays a crucial role in preventing Latent Rehearsal Decay, as it avoids sudden drops in performance and yields higher Final Accuracy. When the Overlap loss is combined with FIFO or Reservoir, it does not consistently improve performance. Specifically, it does not help FIFO converge faster, but on CIFAR it helps Reservoir avoid more severe Latent Rehearsal Decay. Unfortunately the same cannot be said for ImageNet, proving that the Overlap Loss is dependent on maintaining a Deviation-Aware buffer, as it also exploits high-deviation samples in the top-$K$ selection. We hypothesize that FIFO and Reservoir strategies create conditions under which the Overlap loss cannot operate as intended. Reservoir maintains a small subset of well learned samples with low Deviation. Iteratively enforcing low Overlap over an already learned fixed set could be redundant and even amplify the negative effects of Reservoir in the case of a more complex dataset such as ImageNet100. Instead, FIFO retains only the most recent samples, which are mostly from the same task. Consequently, the Overlap loss mostly pushes apart intra-task examples and does not meaningfully reduce overlap across diverse samples, limiting its effectiveness.

## 6.3 INATURALIST - A REAL WORLD SCENARIO STARTING FROM A PRETRAINED NETWORK

We experiment on iNaturalist (Van Horn et al., 2018), a large dataset with natural images and hierarchical labels, to simulate a real world scenario with long OCSSL training runs starting from a pretrained network. Specifically, we pretrained a ResNet-18 backbone with Sim-

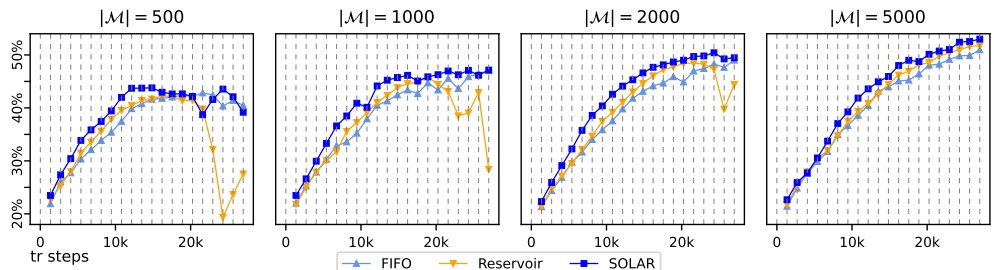

Figure 7: Training curves across buffer sizes $|\mathcal{M}|$ on CIFAR-100. Smaller buffer corresponds to stronger Latent Rehearsal Decay. SOLAR beats FIFO and Reservoir across the training stream.

Siam, using multi-epoch offline training on the *Plantae* category. We employed the pretrained model as basis for an OCSSL scenario instead of starting from scratch, using the the the *Animalia* iNaturalist category for stream training and evaluation. Each task in the OCSSL stream corresponds to each of *Class* categories present in *Animalia*, while 152 different *Order* categories are used as labels for probing, resulting in unbalanced tasks each containing similar classes. This setup allowed us to simulate a real-world scenario, with large distribution shifts and a pretrained network – whose performance we would like to improve – already available.

Table 2: iNaturalist results.

| METHOD | FINAL ACC. | AVG. ACC. |
|---|---|---|
| *pretrained* | 36.42 | – |
| Reservoir | 42.67 | 40.45 |
| FIFO | 43.91 | 40.77 |
| SOLAR | **44.11** | **41.73** |

This setup is close to Continual Pretraining (Cossu et al., 2024), but online. Results are shown in Table 2. All methods show a significant improvement over the direct use of the pretrained baseline, showing the usefulness of continuing pretraining even in an online setting. SOLAR achieves better accuracy than FIFO and Reservoir, once again confirming the soundness of our method. Reservoir achieves a similar Average Accuracy to FIFO but lower Final Accuracy. This is coherent with our previous observations in which Reservoir performance are inferior especially in Final Accuracy due to its lack of plasticity.

## 7 CONCLUSIONS

In this paper we studied continual learning in online, self-supervised settings (OCSSL).We showed that OCSSL induces qualitatively different learning dynamics favoring plasticity over stability. This leads to counterintuitive outcomes, such as FIFO outperforming Reservoir sampling. We explained this through the *Latent Rehearsal Decay* hypothesis, which at-

Table 3: Ablation study on buffer type and effect of the Overlap Loss $\mathcal{L}_{\text{overlap}}$.

| BUFFER | $\mathcal{L}_{\text{overlap}}$ | CIFAR-100 FINAL/AVG. | ImageNet100 FINAL/AVG. |
|---|---|---|---|
| Reservoir | ✗ | 44.4/39.6 | 40.5/39.3 |
| Reservoir | ✓ | 46.3/40.3 | 35.8/37.6 |
| FIFO | ✗ | 49.0/39.3 | 48.7/38.9 |
| FIFO | ✓ | 46.9/38.4 | 45.4/37.9 |
| Deviation-Aware | ✗ | 47.7/41.4 | 46.9/**42.8** |
| Deviation-Aware | ✓ | **49.5/42.3** | 49.4/**42.8** |

tributes performance drops to latent space degradation when replay buffers are small and static. To quantify this effect, we introduced *Deviation* and *Overlap*, two metrics that measure latent degradation and serve as early indicators of probing accuracy decay. Building on these insights, we developed *SOLAR*, which leverages efficient proxies for these metrics to manage replay buffers and preserve latent structure. Experiments demonstrate that *SOLAR* achieves state-of-the-art results on OCSSL computer vision benchmarks, balancing fast convergence (high Average Accuracy) with strong performance (high Final Accuracy), whereas other methods typically trade off one for the other. We hope this work encourages a shift of focus from preventing forgetting to continuously improving the quality of latent representations in OCSSL.

**Limitations and Future works.** In this work we only examined the latent space from an *unsupervised* perspective, without considering metrics of representation space quality that consider task or class labels. As future work, we aim to deepen the study of buffer behavior under large domain shifts when using pre-trained models for downstream tasks. We also hypothesize that Latent Rehearsal Decay may arise at the feature level in *supervised* Online CL, opening another avenue for future exploration.

## 8 REPRODUCIBILITY STATEMENT

We have taken steps to ensure the reproducibility of our work. The full source code, along with scripts to reproduce all results in the paper, will be published after the review period. All experiments were performed on publicly available datasets, and details of model architectures, and main training hyper-parameters are given in the main paper with additional details included in the supplementary material (Appendix C.4 and D.1). To ensure the reproducibility of stochastic processes, such as weight initialization and dataset shuffling, we fix random seeds across all experiments reporting the standard deviations. The random seed values will be clearly documented in our published code.

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

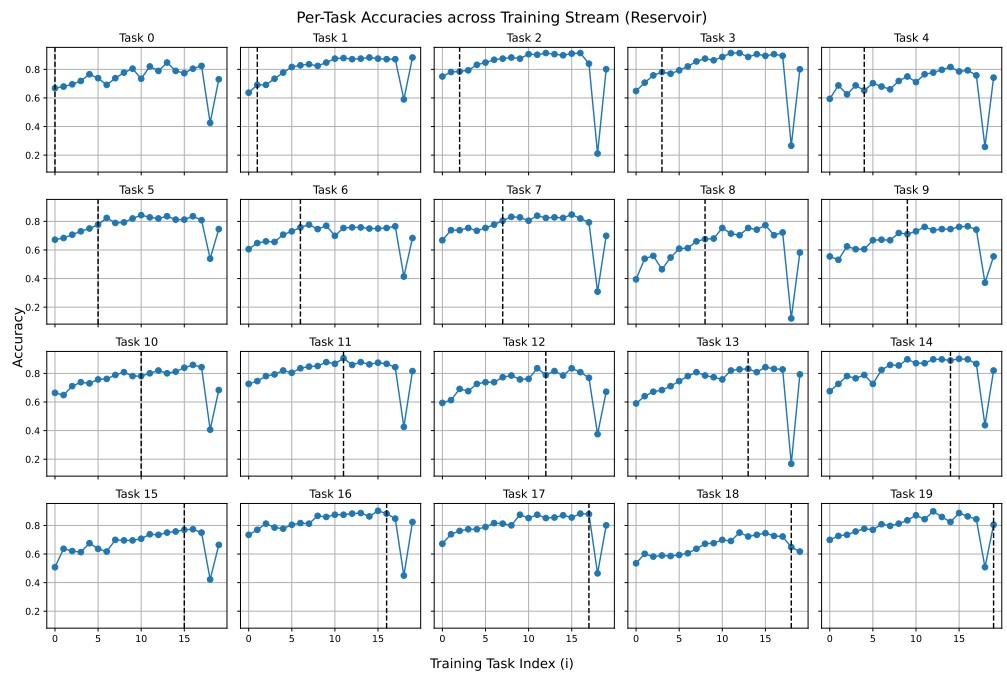

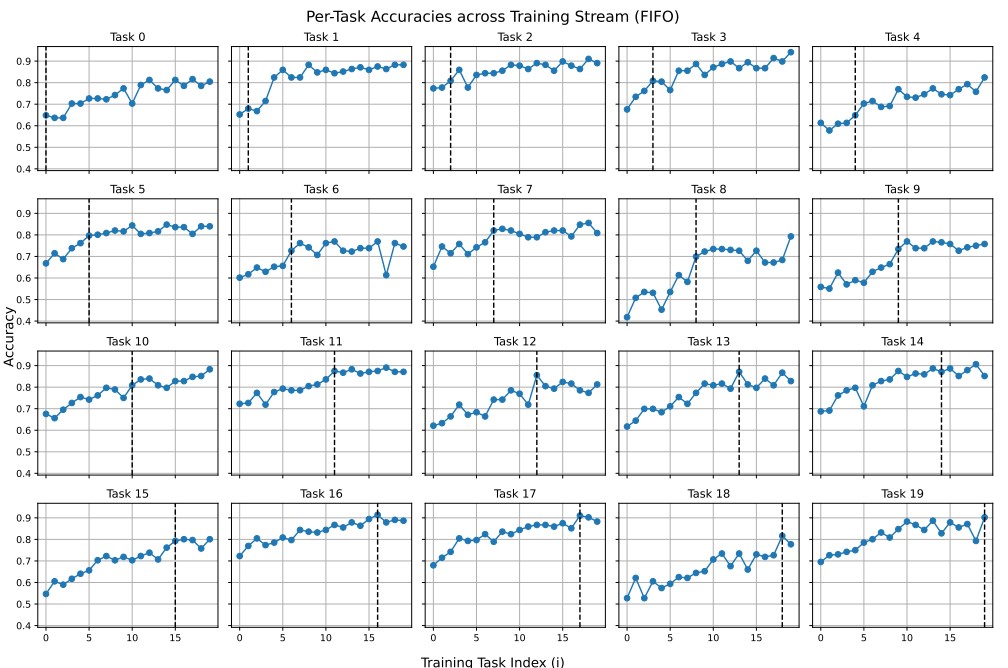

Figure 8: Probing accuracy of each task at the end of each experience in the OCSSL stream (CIFAR-100) for Reservoir (top) and FIFO (bottom). The vertical dashed line indicates the point when the corresponding task just ended in the training stream.

## APPENDIX A  PER-TASK ACCURACY FOR FIFO AND RESERVOIR BUFFERS

Figure 8 reports the probing accuracy curves for every task calculated at the end of each encountered task across CIFAR-100 training stream, for both FIFO and Reservoir. We observe for all tasks an ascending behavior in terms of accuracy, meaning that there is high cross-task transfer in this

scenario, even when the model employs a FIFO buffer and thus rehearsal of earlier tasks is very limited. Nonetheless, and contrary to other papers (Hess et al., 2024), we do not consistently observe instances of feature forgetting: the highest accuracy for each task is often not reached after just finishing training said task.

## APPENDIX B    ADDITIONAL ANALYSIS OF LATENT METRICS

Here we show the relationship between the training loss and Latent Rehearsal Decay. We then complement the figure in the main paper (Figure 3) with an analysis of Latent Rehearsal Decay on CIFAR-100. Finally, we discuss other metrics proposed in the literature for measuring feature collapse or degradation and show that these phenomena are unrelated to Latent Rehearsal Decay in OCSSL.

### B.1    RELATIONSHIP BETWEEN TRAINING LOSS AND LATENT REHEARSAL DECAY

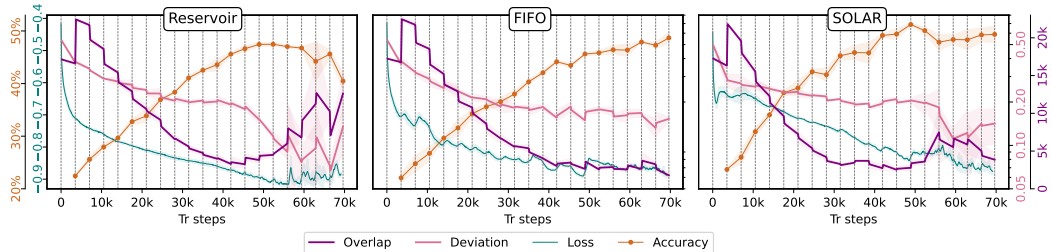

Figure 9: The Figure shows Deviation, Overlap probing accuracy, and smoothed training SSL loss during OCSSL training on ImageNet100. We observe loss instabilities in correspondence with degradation of our metrics, especially for Reservoir.

Figure 9 relates the Self-Supervised training loss to probing accuracy and the Deviation and Overlap metrics on ImageNet100. We see that Reservoir has a flat loss curve, as the buffer composition changes slowly with time and is focused on stability. In later training phases, in correspondence with increasing Overlap and drop in Deviation, we suddenly switch to an instability phase for the loss. This is an indicator that, upon latent degradation, the training process is also disrupted and falls into an instable state. FIFO instead maintains a curve that is not smooth compared to Reservoir, caused by the continually shifting distribution inside the buffer. At the same time it does not suffer from loss instability.

SOLAR maintains a higher training loss across the training process; this is evidence of the effect of the Deviation-Aware buffer that prioritizes high-loss samples during training. The loss curve is not as smooth as Reservoir, indicating that the internal buffer distribution changes more frequently and can thus incorporate plasticity. Nonetheless, SOLAR suffers from slight loss instability at the end of training, in correspondence with minor degradation of latent metrics in later training stages.

### B.2    LATENT REHEARSAL DECAY METRICS ON CIFAR-100

Figure 10 reports the accuracy, Deviation, and Overlap metrics calculated across OCSSL training on CIFAR-100, for both Reservoir and FIFO. We see behavior similar to ImageNet (see Fig. 3), with both methods improving continuously during the initial *Convergence phase*, with decreasing Overlap, and slight decrease in Deviation. Again, the behavior of the two methods diverges in later training. Reservoir suffers from *Latent Rehearsal Decay*, which we observe as a sudden drop of Deviation and increase in Overlap. The degradation of the latent space is followed by a drop in the probing accuracy. Instead, FIFO continuously improves (*Plasticity Learning phase*) in probing accuracy and does not exhibit degradation of the latent space. However, differently from ImageNet, FIFO suffers an increase in Overlap in later phases, but still inferior to Reservoir's increase in Overlap, and it is not accompanied by drops in accuracy or Deviation. This demonstrates that both metrics are required to degrade in order for performance to fall off and fall into Latent Rehearsal Decay.

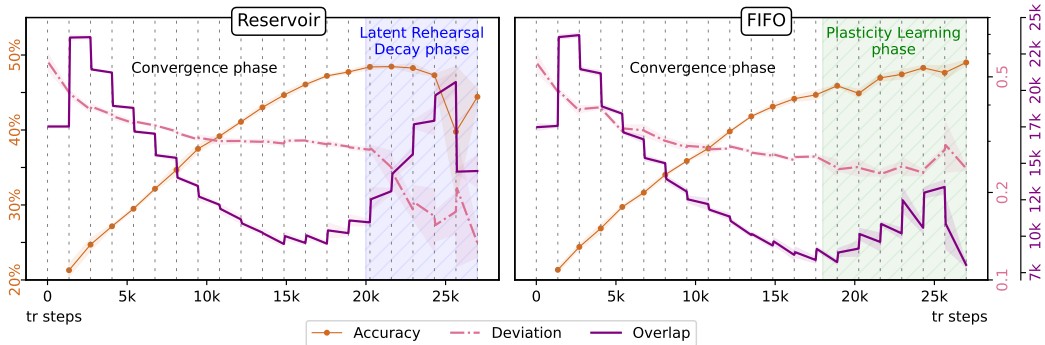

Figure 10: Accuracy and Latent Rehearsal Decay metrics on CIFAR-100. Both FIFO and Reservoir have an initial *Convergence phase*, in which Accuracy increases and both metrics improve. After a certain point, Reservoir transitions into the *Latent Rehearsal Decay phase*, in which the sudden drop in accuracy is preceded by a sharp decrease in Deviation and increase in Overlap.

## B.3 EXISTING METRICS FOR FEATURE DEGRADATION

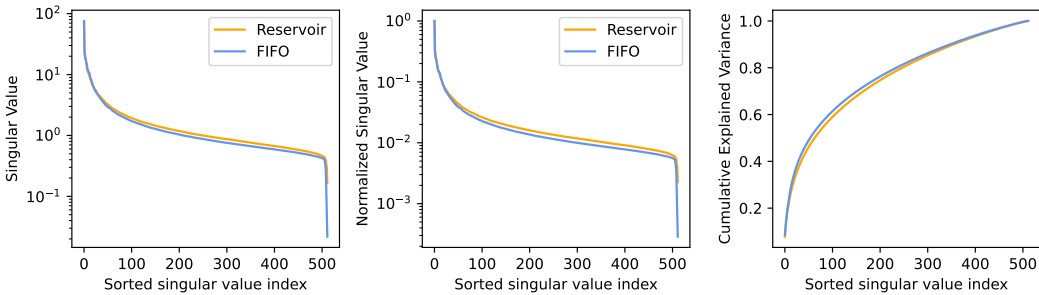

Figure 11: Plot showing no evidence of feature collapse for Reservoir on the CIFAR-100 test set expressed using the metrics based on analyzing SVD decompositions of feature representations (Li et al., 2022).

Prior work links collapse in SSL to feature collapse (Li et al., 2022) or lack of uniformity in the latent space (Wang & Isola, 2020). In this section, we show that these phenomena are unrelated to Latent Rehearsal Decay in OCSSL.

**Feature Collapse.** Feature collapse in non-contrastive SSL methods (such as SimSiam) has been characterized by Li et al. (2022) through the analysis of the singular value decomposition of feature representations. Following their methodology, we report three metrics. Given a feature matrix $A \in \mathbb{R}^{N \times d}$, we first normalize each row to unit norm and compute its singular values $\{\sigma_i\}_{i=1}^{d}$. (1) *Singular value spectrum:* the decay of $\{\sigma_i\}$ as a function of their sorted index, which reflects the effective dimensionality of the learned representation. (2) *Normalized singular value spectrum:* the relative distribution of singular values, defined as $\tilde{\sigma}_i = \sigma_i / \sigma_1$, where $\sigma_1$ is the largest singular value. This metric shows how balanced or uneven the spread of information is across different feature directions. (3) *Cumulative explained variance:* the fraction of variance captured by the top-$k$ singular values, defined as $\text{CEV}(k) = \sum_{i=1}^{k} \sigma_i / \sum_{j=1}^{d} \sigma_j$, which measures how quickly the singular values concentrate, with steeper curves indicating stronger collapse into a low-dimensional subspace.

Figure 11 reports these three metrics on the entire test set features at the end of CIFAR-100 training for both Reservoir and FIFO. The two methods produce almost identical results, despite Reservoir having already exhibited Latent Rehearsal Decay. This indicates that Latent Rehearsal Decay is not related to feature collapse.

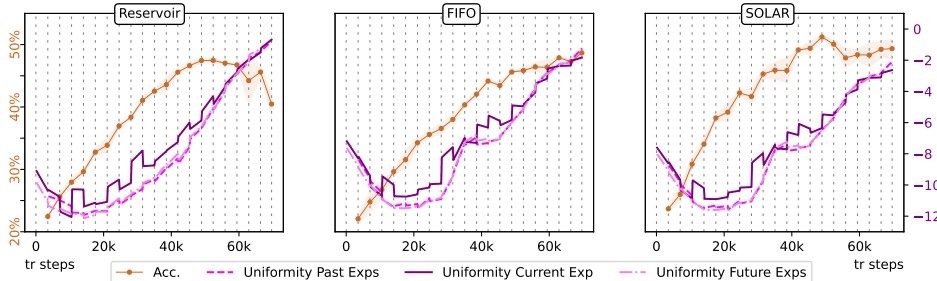

Figure 12: Uniformity loss, a measure of SSL collapse (Wang & Isola, 2020), calculated on past, future and current experiences for Reservoir, FIFO and SOLAR on ImageNet100.

We conduct a more fine-grained analysis of the singular value spectra of test features, extending previous results in Figure 13. Specifically, we report SVD-based feature metrics for both FIFO and Reservoir replay at three points during training – early, mid, and late – and we further disaggregate these metrics by whether the evaluated data originate from past, current, or future experiences relative to the training step. Across all settings, FIFO and Reservoir exhibit nearly identical plots on all three SVD metrics. The only notable discrepancy arises at the end of training, where FIFO shows a slightly stronger collapse in the last single singular direction; this effect appears mildly attenuated for Reservoir. Interestingly, the SVD curves for past and future experiences are nearly indistinguishable, suggesting that training on a particular distribution does not induce asymmetric distortions in the learned feature space. This observation is consistent with our OCSSL forgetting analysis, where future-task performance improves at the same pace of past-task performance (see Section 2). Even with this more detailed breakdown, we find no evidence linking feature collapse to Latent Rehearsal Decay.

Finally, note that the tail of the singular value spectrum for the current experience is consistently lower than that of the aggregated past or future experiences. This is expected: the current task contains a narrower sample distribution, and thus some feature directions naturally exhibit lower variability within the current experience while remaining informative when considering broader past or future distributions.

**Uniformity Loss.** The uniformity loss, introduced by Wang & Isola (2020) as a measure of representation space quality, penalizes feature vectors that are not uniformly distributed on the latent hypersphere. Figure 12 correlates the uniformity loss to probing accuracy across OCSSL training on ImageNet100. After an initial drop in the loss all three methods (Reservoir, FIFO and SOLAR) show a gradual and constant increase of uniformity loss, with a similar behavior. The only difference that can be observed is slightly higher final uniformity loss for Reservoir, which may be weakly related to concurrent Latent Rehearsal Decay.

Prior work links collapse in SSL to feature collapse (Li et al., 2022) or lack of uniformity in the latent space (Wang & Isola, 2020). In this section, we show that these phenomena are unrelated to Latent Rehearsal Decay in OCSSL.

### B.4 Latent Rehearsal Decay vs Buffer Overfitting

We argue that Latent Rehearsal Decay is fundamentally different from the well-studied notion of memory overfitting in supervised CL (Zhang et al., 2022; Khan et al., 2024). In particular, Yan et al. (2024) characterizes buffer overfitting as the relative performance gap between buffer and test accuracy, implicitly assuming that representations of buffer samples remain strong—with clear class boundaries—while representations for out-of-buffer samples deteriorate. Under this view, one would expect metric degradation to occur only for data outside the buffer. However, Figure 14 shows metric trajectories computed both on the full training set and on buffer samples alone, and the curves are highly aligned. For Reservoir, we observe a decrease in Deviation for both memory and non-memory samples; notably, the rise in overlap occurs in both subsets, indicating that sample discernibility declines even for the ostensibly overfit buffer items. We hypothesize that buffer overfitting acts as

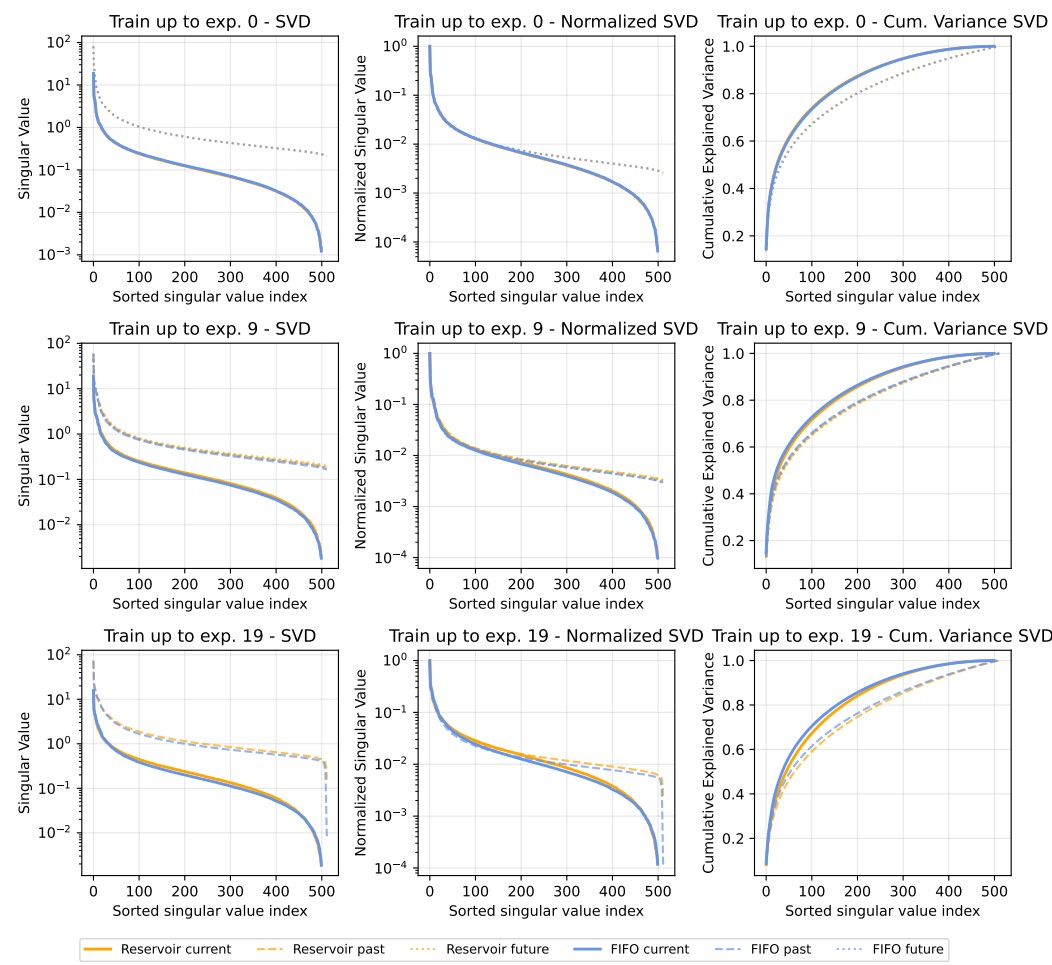

Figure 13: Figure reports metrics based on analyzing SVD decompositions of feature representations (Li et al., 2022) for Reservoir and FIFO. Features are extracted at the beginning, middle and end of OCSSL training on CIFAR100 (each row of plots shows metrics for each training checkpoint). We include lines for features SVD on current, past and future experiences, relative to the training checkpoint experience.

a contributing cause of Latent Rehearsal Decay, but it does not fully describes for the collapse we observe.

## APPENDIX C    ADDITIONAL SOLAR DETAILS

In this appendix we provide the full pseudocode of SOLAR. We show the relationship between the self-supervised loss and the Deviation metric, motivating the use of the SSL loss in SOLAR as an approximation of Deviation. Finally, we compare the true Overlap – which requires multiple backward passes and is infeasible in OCSSL – with the Online Overlap estimation employed by SOLAR, and we ablate the hyperparameter associated with the Overlap loss.

### C.1    SOLAR ALGORITHM PSEUDO-CODE

We provide the full pseudo-code of SOLAR in Algorithm 1 below. The algorithm consists of an EXTRACT function, which retrieves samples $x$ from the Deviation-Aware buffer (accumulated in a previous time step via the UPDATE function), and forwards them through the backbone. The overlap loss $\mathcal{L}_{\text{overlap}}$ and the SSL loss $\ell$ are then computed, followed by backpropagation. Note that the

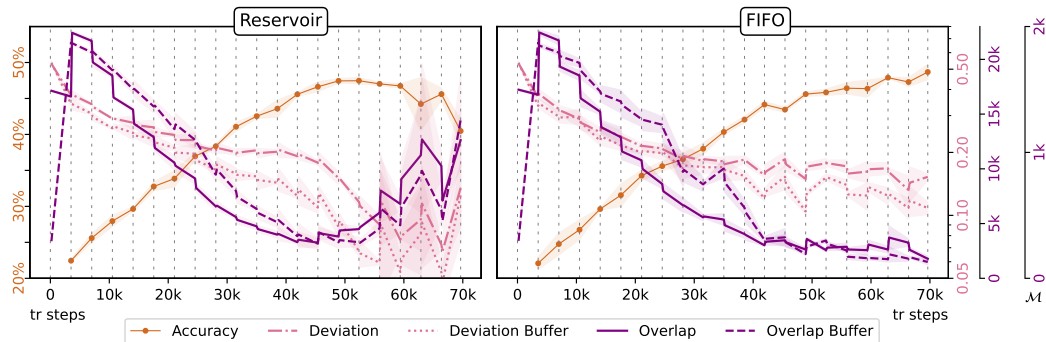

Figure 14: Probing accuracy, Deviation and Overlap metrics calculated on the entire training stream and only on buffer samples (training on ImageNet100), for both Reservoir and FIFO. We observe similar curves for metrics on the buffer and on the entire training data.

overlap loss is computed using as targets, the average features $\hat{\mathbf{z}}$ and average angles $\hat{\theta}$ associated to the top-k SSL losses stored in the buffer (always via the UPDATE function).

The UPDATE function is called after each backpropagation step to store current samples $x$, the current SSL loss $\ell$, the average angle $\hat{\theta}$, and the average feature representation $\hat{\mathbf{z}}$ in the Deviation Aware buffer. These are updated online for each sample $j$ via an exponential moving average (EMA), resulting in $l_j^{\mathcal{M}}, \bar{\mathbf{z}}_j^{\mathcal{M}}, \bar{\theta}_j^{\mathcal{M}}$. Note that higher loss samples are discarded when the maximum buffer size is reached.

### C.2 LOSS AS A PROXY FOR DEVIATION

In this section we demonstrate that the per-sample self-supervised loss is a good proxy for estimating Deviation, as the two are positively related by:

$$\boxed{\frac{d\,\mathrm{Dev}}{d\mathcal{L}_{SSL}} \;=\; \frac{1}{n^2} \;>\; 0\,.}$$

**Setup and Assumptions.** Let us assume a generic (positive-only) SSL instance discrimination loss that tries to minimize the generic similarity $S$ among feature views $\mathbf{z}^i$:

$$\mathcal{L}_{SSL} \;=\; -\sum_{i \neq j} S(\mathbf{z}^i, \mathbf{z}^j). \tag{8}$$

It is a reasonable assumption to consider $S$ to be positively related to cosine similarity $S_C$, thus, for simplicity, we set $S = S_C$.

Recall the definition of *Deviation* from equation 1:

$$\mathrm{Dev}(\mathbb{T}_a) = \frac{1}{|\mathbb{T}_a|^2} \sum_{\mathbf{z}_a^i, \mathbf{z}_a^j \in \mathbb{T}_a} \left(1 - S_C(\mathbf{z}_a^i, \mathbf{z}_a^j)\right).$$

Now we rewrite this formulation by sampling a fixed number of views, $n$, from $\mathbb{T}_a$:

$$\mathrm{Dev} \;=\; \frac{1}{n^2} \sum_{i=1}^n \sum_{j=1}^n \left(1 - S_C(\mathbf{z}^i, \mathbf{z}^j)\right) \;=\; 1 - \underbrace{\frac{1}{n^2} \sum_{i=1}^n \sum_{j=1}^n S_C(\mathbf{z}^i, \mathbf{z}^j)}_{=:\bar{S}}, \tag{9}$$

where $\bar{S}$ is the average (including self-pairs) cosine similarity.

**Relating $\mathcal{L}_{SSL}$ and Dev.** We can rewrite equation 8 in terms of the double sum in equation 9:

$$\mathcal{L}_{SSL} = -\sum_{i=1}^n \sum_{j=1}^n S_C(\mathbf{z}^i, \mathbf{z}^j) \;+\; \sum_{i=1}^n S_C(\mathbf{z}^i, \mathbf{z}^i) = -n^2 \bar{S} \;+\; n\,,$$

---

**Algorithm 1** Pseudo-code for the SOLAR training loop

**Given:** SSL model $f$, total batch size $B_{tot}$, buffer $\mathcal{M}$ composed by entries $\langle x_i^{\mathcal{M}}, \ell_i^{\mathcal{M}}, \bar{\mathbf{z}}_i^{\mathcal{M}}, \bar{\theta}_i^{\mathcal{M}}, e_i^{\mathcal{M}} \rangle$, buffer maximum size $M$.

```
 1: for b_s in D do                                          ▷ streaming minibatches
 2:     for p in n_p do
 3:         if p == 0 then                 ▷ if first minibatch pass train also on stream batch b_s
 4:             x ← EXTRACT(M, B_tot − |b_s|) ∪ b_s
 5:         else                                      ▷ otherwise train only using M samples
 6:             x ← EXTRACT(M, B_tot)
 7:         end if
 8:         x¹, x² ← Augmentations(x)
 9:         z¹, z² ← f(x¹), f(x²)
10:         ℓ ← L_SSL(z¹, z²)                   ▷ ℓ = [ℓ_i]_{i=1}^{B_tot} is the per sample SSL loss
11:         Select K samples from M \ x, with highest ℓ^M → [T_K^M = ⟨z̄_k^M, θ̄_k^M⟩]_{k=1}^K
12:         L_overlap = (1/B_tot) Σ_{i=1}^{B_tot} (1/K) Σ_{k=1}^K max(0, Ov(T_i, T_k^M))
13:         L = ℓ.mean() + ωL_overlap
14:         BACKPROP(L)
15:         UPDATE(M, x, ℓ, ẑ, θ̂)
16:     end for
17: end for
```

```
18: function UPDATE(M, x, ℓ, ẑ, θ̂)
19:     for x_i ∈ x do
20:         if x_i ∈ M in position j then
21:             ℓ_j^M ← 0.5 · ℓ_j^M + 0.5 · ℓ_i
22:             z̄_j^M ← 0.5 · z̄_j^M + 0.5 · z̄_i
23:             θ̄_j^M ← 0.5 · θ̄_j^M + 0.5 · θ̄_i      ▷ EMA update for samples already in buffer
24:         else
25:             M.append(⟨x_i, ℓ_i, ẑ_i, θ̂_i, 0⟩)                    ▷ append for new samples
26:         end if
27:     end for
28:     if |M| > M then
29:         r ← |M| − M                            ▷ r is the number of samples to remove
30:         Sort M in ascending order by ℓ^M              ▷ remove lowest loss samples
31:         Remove the first r samples from M
32:     end if
33: end function
```

```
34: function EXTRACT(M, b)                         ▷ b is the number of samples to extract
35:     ē ← MinMaxNormalization(e^M)
36:     p ← 1 − Softmax(ē)
37:     I ← ∅                                         ▷ initialize extraction set
38:     for j = 1 to b do                             ▷ sampling without replacement
39:         Sample index i from {1, ..., N} \ I with probability p_i
40:         I ← I ∪ i
41:         e_i^M ← e_i^M + 1                        ▷ increase extraction count for sample i
42:     end for
43:     return [x_i^M]_{i∈I}
44: end function
```

---

because $S_C(\mathbf{z}^i, \mathbf{z}^i) = 1$ for every $i$. Thus we obtain:

$$n^2 \bar{S} \;=\; n - \mathcal{L}_{SSL}.$$

Plug this into equation 9 (recall $\mathrm{Dev} = 1 - \bar{S}$) to obtain

$$\mathrm{Dev} \;=\; 1 - \frac{n - \mathcal{L}_{SSL}}{n^2} \;=\; \frac{\mathcal{L}_{SSL}}{n^2} + \frac{n^2 - n}{n^2} \;=\; \frac{\mathcal{L}_{SSL}}{n^2} + \frac{n-1}{n}.$$

**Conclusion.** The preceding identity shows that $\mathrm{Dev}$ is an affine (linear + constant) function of the loss $\mathcal{L}_{SSL}$:

$$\mathrm{Dev} \;=\; \frac{1}{n^2}\,\mathcal{L}_{SSL} \;+\; \frac{n-1}{n}\;.$$

Hence we obtain our initial statement:

$$\frac{d\,\mathrm{Dev}}{d\mathcal{L}_{SSL}} \;=\; \frac{1}{n^2} \;>\; 0,$$

thus proving that $\mathrm{Dev}$ is *positively related* to $\mathcal{L}_{SSL}$: increasing $\mathcal{L}_{SSL}$ increases $\mathrm{Dev}$, and decreasing $\mathcal{L}_{SSL}$ decreases $\mathrm{Dev}$. In particular, minimizing the loss $\mathcal{L}_{SSL}$ during training reduces the Deviation metric.

### C.3 ONLINE ESTIMATION OF OVERLAP

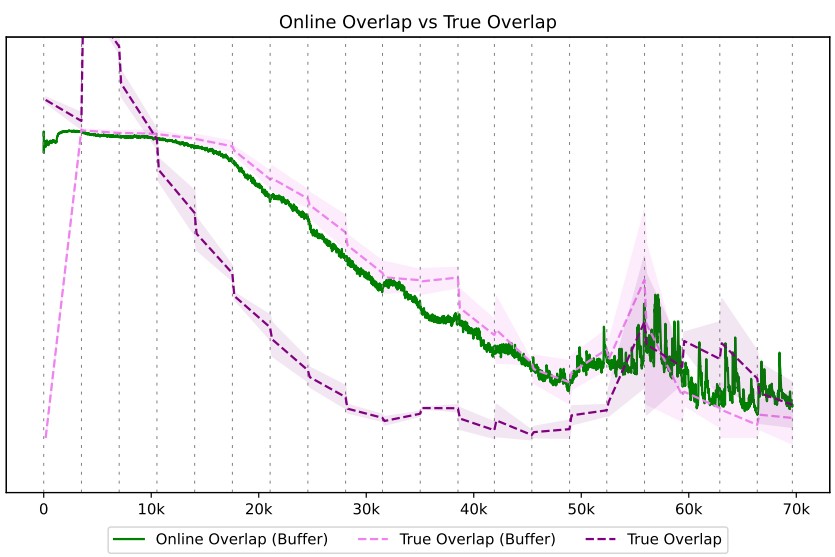

Figure 15: Online versus true Overlap calculated for SOLAR on ImageNet100, with $|\mathcal{M}| = 2000$. Online Overlap estimation is a good proxy for the *true Overlap*. Online Overlap closely matches the true Overlap calculated on the buffer only, demonstrating that the slight mismatch is only due to the buffer not being representative enough of the entire stream.

We compare the true Overlap – which requires multiple forward passes and is infeasible in OCSSL – with the Online Overlap estimation employed by SOLAR. Figure 15 shows the *true Overlap*, the true overlap calculated on the buffer only – both calculated offline with a forward pass on the data – and the online Overlap, which is again calculated during training only on the buffer, employing $\bar{z}_i^t$ and $\bar{\theta}_i^t$ extracted during training.

Online Overlap estimation serves as a reliable proxy for the *true Overlap*. In practice, we observe that Online Overlap closely matches the ground-truth Overlap computed over the buffer only. We do observe some deviation between the true Overlap and its online estimation, though both exhibit the same first-order trends. These discrepancies can be largely attributed to the buffer's limited representativeness of the entire data stream, rather than to any intrinsic weakness of the estimation procedure itself. This distinction is important: it suggests that the quality of the buffer, rather than the quality of the estimator, ultimately governs the accuracy of the measurement.

## C.4 ABLATION ON THE $\omega$ HYPERPARAMETER FOR $\mathcal{L}_{\text{OVERLAP}}$

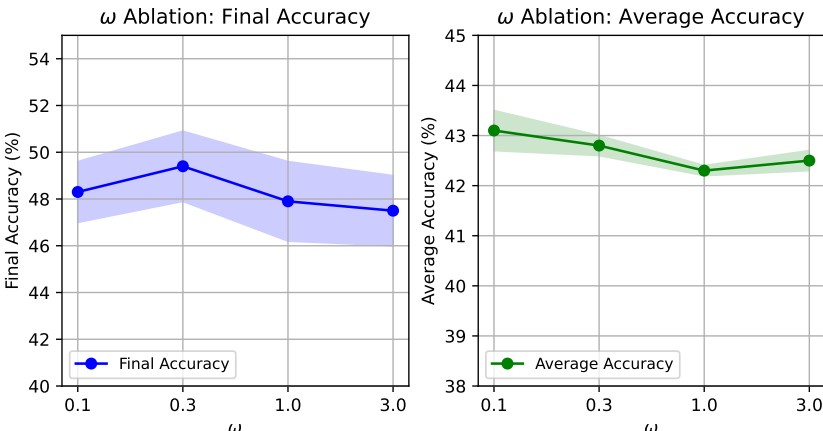

Figure 16: Ablation on $\omega$, the hyperparameter controlling overlap loss $\mathcal{M}_{\text{overlap}}$ strength. The mean and standard deviation of Final and Average Accuracy for ImageNet100 are reported.

Figure 16 presents an ablation study on the hyperparameter $\omega$, which controls the strength of the overlap loss $\mathcal{L}_{\text{overlap}}$ in SOLAR. Both Average and Final Accuracy on ImageNet100 are reported. We observe that $\omega$ has a limited effect on performance, indicating that SOLAR is relatively insensitive to the choice of this hyperparameter – an advantageous property in online settings such as OCSSL. Overall, slightly lower Final Accuracy occur at smallest values of $\omega$, consistent with the ablation results in Table 3. This behavior aligns with expectations, as lower $\omega$ makes SOLAR closer to just the Deviation Aware buffer, which performs worse than the full SOLAR formulation in Final Accuracy.

We use $\omega = 0.3$ only for the main experiments on ImageNet100, for all other setups, we used a fixed $\omega = 1.0$.

## C.5 ADDITIONAL ABLATIONS ON SOLAR

Table 4: Ablation study on effect of extraction criterion (left) and use of EMA to update buffer metrics (right).

| EXTRACTION | CIFAR-100 FINAL/AVG. | ImageNet100 FINAL/AVG. |
|---|---|---|
| SOLAR + random extraction | 48.7/42.2 | 43.3/42.0 |
| SOLAR + $\ell_i^{\mathcal{M}}$ extraction | 49.4/42.0 | 41.9/42.3 |
| SOLAR + $e_i^{\mathcal{M}}$ extraction | **49.5/42.3** | **49.4/42.8** |

| | EMA $\mathcal{M}$ STATISTICS | CIFAR-100 FINAL/AVG. | ImageNet100 FINAL/AVG. |
|---|---|---|---|
| SOLAR | ✗ | 49.0/41.3 | **49.4**/42.0 |
| SOLAR | ✓ | **49.5/42.3** | **49.4/42.8** |

Table 4 analyzes the effects of different sample extraction strategies in the Deviation Aware Buffer (left) and the use of EMA-updated metrics for updating online buffer statistics (namely $\ell_i^{\mathcal{M}}$, $\bar{z}_i^{\mathcal{M}}$, $\bar{\theta}_i^{\mathcal{M}}$). On CIFAR-100, the extraction strategy has minimal impact, with a slight advantage for using softmax of the extraction count ($e_i^{\mathcal{M}}$). Instead, on ImageNet100, using random or extraction based on the softmax of normalized per-sample loss ($\ell_i^{\mathcal{M}}$) yields far inferior results in final accuracy. This demonstrates the need for this element in our method, especially for more complex datasets such as ImageNet in which an additional component is needed to enhance diversity and thus Deviation. In particular, we hypothesize that the loss-based extraction is redundant with the loss-based criterion for sample storage in the buffer, causing too strong of a bias towards high-loss samples during training, which negatively impacts diversity.

Instead, the use of EMA metrics has limited impact on Final Accuracy, favoring more Average Accuracy. Coherently with previous studies (Cignoni et al., 2025b; Purushwalkam et al., 2022), the use of EMA statistics enhances their reliability across all stream training.

## C.6 COMPARISON WITH OTHER METHODS PRIORITIZING "HARD" BUFFER SAMPLES.

Table 5: Comparison of components of methods prioritizing "hard" samples in memory replay.

| METHOD | UPDATE POLICY | EXTRACT POLICY | COMPUTATIONAL SCALABILITY |
|---|---|---|---|
| MIR | Reservoir | Max loss interference | ✗ |
| GSS | Max gradient diversity | Random | ✗ |
| PER | FIFO | Soft priority on highest loss | ✓ |
| LARS | Reservoir + lowest-loss elimination | Random | ✓ |
| Deviation-Aware | Lowest loss | Soft priority on lowest $e_i^{\mathcal{M}}$ | ✓ |

Prioritizing "hard" samples for memory replay has been widely studied in the literature (Schaul et al., 2015), including in CL (Aljundi et al., 2019b;a; Buzzega et al., 2021). MIR (Aljundi et al., 2019a) replays buffer samples that induce the largest loss interference, while GSS (Aljundi et al., 2019b) maintains a buffer of samples with maximally diverse gradients. Although effective, both approaches are prohibitively expensive in the OCSSL setting: MIR requires an additional forward pass over the entire buffer at each step, and GSS repeatedly estimates gradients for buffer samples at each step.

A seemingly more efficient alternative is PER (Schaul et al., 2015), which uses a FIFO buffer combined with prioritized sampling based on TD-error. In OCSSL, this mechanism can be instantiated by replacing TD-error with SSL loss. However, our Deviation-Aware buffer fundamentally differs from PER. PER biases only the extraction policy toward hard samples, while leaving the buffer update mechanism untouched—thus failing to control which samples populate the buffer over time. As a result, PER performs on par with FIFO in our experiments (Table 1), underscoring the importance of managing the buffer composition rather than only its sampling distribution.

LARS (Buzzega et al., 2021) also adopts a loss-based criterion, but only for deciding which sample to evict upon insertion. Despite this modification, its dynamics remain close to Reservoir: insertion probabilities monotonically shrink, causing the buffer to converge toward a fixed subset. Consequently, its performance closely mirrors that of Reservoir (Table 1).

In contrast, our Deviation-Aware buffer jointly addresses both update and extraction policies, explicitly promoting hard samples while maintaining diversity throughout training. This design is grounded in OCSSL-specific insights—namely, our new deviation metrics and the Latent Rehearsal Decay analysis—which together provide principled motivation for the proposed approach.

Table 5 summarizes the key differences among the methods reviewed in this section.

## APPENDIX D    MORE ON EXPERIMENTS

In this appendix we provide additional details on the experimental setup used in the main paper, analyze the distillation regularization employed by CLA-R and its relationship to Latent Rehearsal Decay, and extend the analysis of buffer size introduced in the main paper with experiments on ImageNet100.

### D.1    ADDITIONAL DETAILS ON EXPERIMENTAL SETUP

**Backbone.** We chose ResNet-18 as backbone, initialized from scratch, as it is a lightweight encoder network, widely tested in CL (Soutif-Cormerais et al., 2023; Urettini & Carta, 2025), and especially in OCSSL (Yu et al., 2023; Cignoni et al., 2025b). As commonly done in the literature, for CIFAR-100 we substituted the first 7x7 convolutional layer with a 3x3 convolutional layer and removed the first MaxPool.

**Optimization.** We employ plain SGD, with momentum = 0.9 and weight decay = 1e-4. We employed a different learning rate for each of the 3 benchmarks, respectively 0.05, 0.02 and 0.01 respectively for CIFAR-100, ImageNet100 and CLEAR100. All methods use these same learning rates for the corresponding benchmarks. All other hyperparameters of the methods were kept fixed as in their original implementation.

**Probing.** Probing is performed with a linear probe trained with a minibatch size of 256 and initial learning rate of 0.05, which decreases by a factor of 3 whenever the validation accuracy stops improving. Training of the probe stops when a minimum learning rate or 100 epochs are reached. We reserve 10% of each training split as validation data.

**Augmentations.** Extraction of multiple views from a single image is performed in two ways: first, to obtain the two views used for the SSL instance discrimination training; secondly, to extract 20 views for calculating overlap and deviation metrics offline. We employ the same set of augmentations for all methods: `RandomCrop`, `RandomHorizontalFlip`, randomly applied `ColorJitter` and `RandomGrayscale`.

**SCALE.** SCALE is the only tested strategy that does not employ SimSiam as the underlying SSL method, as it has its own contrastive loss $\mathcal{L}^{\text{cont}}$. Additionally, the original implementation of the PSA buffer would require a forward pass to calculate features of the buffer that are as recent as possible; this would be not only computationally burdensome, but would also break OCSSL assumptions of lightweight training. For this reason, we estimate buffer features via an EMA instead, similarly as done in SOLAR for $\bar{z}_i^{\mathcal{M}}$; this EMA update of buffer features is exploited also by other OCSSL strategies (Purushwalkam et al., 2022; Cignoni et al., 2025b).

## D.2 FURTHER ANLAYSIS OF CLA

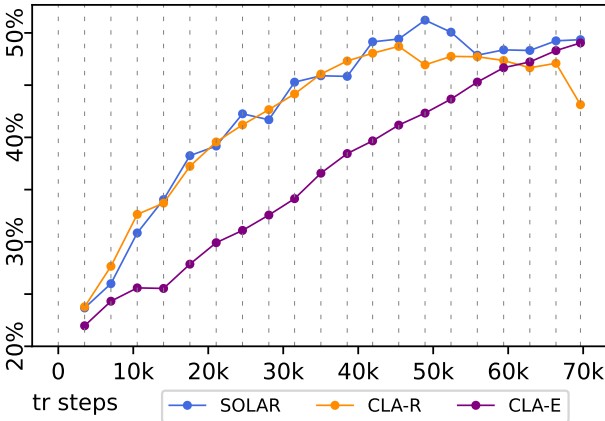

Figure 17: Dynamic regularization versus fixed distillation. These training curves on ImageNet100 show that CLA-E converges slowly when compared to SOLAR, while CLA-R, despite fast convergence, suffers from Latent Rehearsal Decay.

Figure 17 shows the training curves for CLA-E, CLA-R, and SOLAR on ImageNet100. In the early phases, CLA-R achieves accuracy comparable to SOLAR, highlighting its fast convergence. However, towards the end of training, its accuracy drops – imilarly to Reservoir in Figure 3 – a phenomenon that can be attributed to Latent Rehearsal Decay. In contrast, CLA-E exhibits much higher plasticity: its accuracy remains almost flat throughout training, with weaker performance in the initial stages. Nevertheless, akin to FIFO (see Figure 3), its plasticity prevents accuracy degradation over time.

## D.3 CHANGING BUFFER SIZE ON IMAGENET100

Figure 18 reports Final and Average Accuracy on ImageNet100 when the maximum dimension of the buffer $|\mathcal{M}|$ is changed. Similar to CIFAR-100, the final accuracy of Reservoir sampling is strongly affected by a reduced buffer size, whereas FIFO remains comparatively more stable. In contrast to CIFAR-100 (see Figure 5), however, the performance of MinRed also degrades under reduced buffer capacity, suggesting that constructing a maximally representative set of samples is impractical for complex datasets when the available buffer is too small. SOLAR, on the other hand, remains largely unaffected by buffer size reductions and continues to outperform other methods in terms of Average Accuracy.

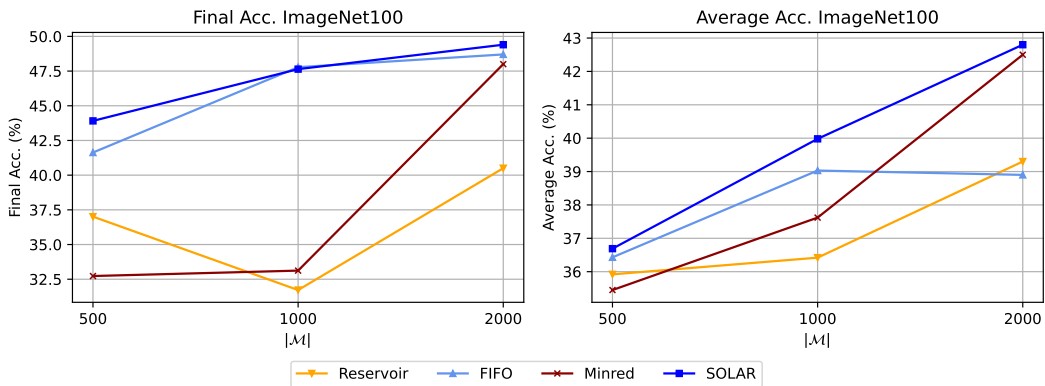

Figure 18: Changing buffer sizes on ImageNet100. We note that MinRed performance is greatly impacted by the reduced buffer size.

## D.4 Training Times

Table 6: Training times (in seconds) of compared methods calculated on a single task on ImageNet100.

| Method | Time ($s$) |
|---|---|
| Reservoir | 2146 |
| FIFO | 2133 |
| MinRed | 2061 |
| LARS | 2150 |
| PER | 2215 |
| SCALE | 4288 |
| CLA-E | 2235 |
| CLA-R | 2126 |
| SOLAR | 2409 |

Table 6 reports the traininig times of different methods calculated on a single task while training on an OCSSL stream of ImageNet100, using the same setup as the main experiments (Sec. 6). We exclude from time calculations the probing and the offline calculation of the metrics, as they are not used in training but only needed for evaluation purposes. All methods exhibit comparable runtimes, with the exception of SCALE, which incurs a substantial overhead due to the PSA buffer. SOLAR requires slightly longer runtime than the other methods, likely because it introduces both an additional loss term and non-trivial buffer policies.

## D.5 SimCLR Experiments

Table 7: Results on streaming online CIFAR-100 (20 experiences) and ImageNet-100 (20 experiences), with SimCLR backbone. Best in **bold**, second best underlined.

| Method | Buffer | Distill. | CIFAR-100 (20 exps) FINAL ACC. | AVG. ACC. | ImageNet100 (20 exps) FINAL ACC. | AVG. ACC. |
|---|---|---|---|---|---|---|
| ER | Reservoir | ✗ | 45.16 | 42.03 | 47.90 | 43.02 |
| ER | FIFO | ✗ | 46.35 | 41.81 | **49.38** | 42.12 |
| MinRed | MinRed | ✗ | 43.92 | 43.05 | 45.54 | 43.30 |
| SCALE | PSA | ✓ | 32.15 | 27.28 | 36.45 | 31.48 |
| SOLAR | Deviation-Aware | ✗ | **46.62** | **43.35** | 49.10 | **43.53** |

We replicated experiments with the main experimental setup (see Section 6), but using Sim-CLR (Chen et al., 2020) instead of SimSiam as the underlying SSL method. Even drastically

changing the SSL loss, and similar insights can be deducted. Again, FIFO scores very well in Final accuracy, but falls slightly behind in Average Accuracy. On the other hand, we still have Reservoir underperforming in Final accuracy but scoring a competitive Average Accuracy.

The effect of Latent Rehearsal Decay for Reservoir seems less damaging, while MinRed instead seems still impacted by it, with lower Final Accuracy. SCALE, again, underfits on both datasets. SOLAR performs best except for Final Accuracy on ImageNet100, falling slightly behind to FIFO. Figure 19 presents probing accuracies during OCSSL training on CIFAR100 with SimCLR. As expected, FIFO is outperformed by Reservoir at the beginning of training, while the situation reverses in later stages and SOLAR stays on top most of the time. Differing from SimSiam, Reservoir does not present a sudden drop of accuracy, but rather a small inflection of the curve. In general, we hypothesize that SimCLR contrastive SSL loss has a similar effect to SOLAR $\mathcal{L}_{overlap}$ as they minimize conceptually similar losses.

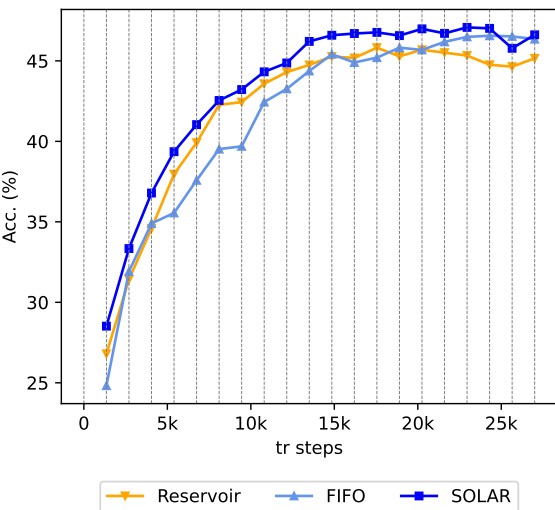

Figure 19: Probing accuracies during training on CIFAR100 with SimCLR.

