# OpenReview forum: "Preventing Latent Reharsal Decay in Online Continual SSL with SOLAR"
_ICLR.cc/2026/Conference — Submitted to ICLR 2026_

### Official Review · Reviewer_fBLe · 2025-10-25

**Soundness:** 2
**Presentation:** 2
**Contribution:** 2
**Rating:** 4
**Confidence:** 4

**Summary:**

This paper investigates the long-term performance degradation issue in Online Continual Self-Supervised Learning (OCSSL), which the authors term "Latent Rehearsal Decay." They propose a replay strategy called SOLAR, which includes a loss-based buffer sampling method (the Deviation-Aware Buffer) and an additional contrastive loss term (Overlap Loss).

**Strengths:**

+ The paper identifies a compelling and counter-intuitive phenomenon in OCSSL: the simple FIFO strategy outperforming the theoretically more stable Reservoir strategy over long training schedules.
+ The experimental validation is thorough, covering multiple variables and validating the method's effectiveness against several baselines.

**Weaknesses:**

1. **Problem Reframing:** The proposed concept of "Latent Rehearsal Decay," while well-analyzed, can be seen as a well-known issue in continual learning—overfitting on the replay buffer. When the buffer's contents become static, the model repeatedly sees the same data, which naturally leads to a decline in generalization. It could be argued that the paper provides a new name for an existing problem rather than discovering a completely new phenomenon.
2. **Incremental Contribution:** The core ideas behind the SOLAR method are not entirely new. Its essence is to prioritize "hard samples" (i.e., those with high loss), a concept with well-established precedents in Experience Replay, notably "Prioritized Experience Replay" (PER). Applying this idea to a self-supervised context, using the SSL loss as a proxy for difficulty, is a relatively direct application and lacks fundamental novelty.
3. **Lack of Deeper Intuition on the Proposed Metrics**: The paper demonstrates that its proposed metrics (Deviation and Overlap) can detect the failure mode while metrics (SVD, uniformity) cannot. However, it **fails to provide a deep, intuitive explanation** for _why_ this is the case. The analysis does not go much beyond showing that "they work," leaving a gap in our fundamental understanding of the phenomenon.
4. **Compatibility with other Self-Supervised Algorithms:** The validation is exclusively on SimSiam (a non-contrastive method). It is unclear how SOLAR's "Overlap Loss" would interact with contrastive methods like SimCLR or MoCo, where it might become redundant or even conflict with the native loss function.
5. **Lack of Quantitative Computational Analysis:** Although the authors considered computational cost in their design, the paper lacks a quantitative experiment comparing the computational overhead (e.g., training time, FLOPs) against other methods.
6. **Incomplete Experimental Analysis**: The analysis of the SVD experiment in the appendix feels incomplete. Unlike the uniformity analysis, it is only performed on the final test set. A more comprehensive analysis across past, present, and future data splits could have provided crucial insights into the dynamics of the feature space, representing a **missed opportunity** to strengthen the paper's core claim.
7. Although I know that Online Continual SSL is not proposed for the first time in this paper, I still have some major doubts about this setting. I feel that pre-training and data streaming are somewhat contradictory concepts: pre-training usually requires sufficient training time and a large amount of data to obtain general representations applicable to most data, whereas data streaming emphasizes fast adaptation and avoiding forgetting. I hope the authors can provide some real-world examples where this setting is applicable.

**Conclusion:** Given the high bar for conceptual and methodological innovation at ICLR, I am inclined to recommend rejection.

**Questions:**

See the weakness

---

> ### Author Response · Authors · 2025-11-20
> **Response 1/2**
>
> We thank the reviewer for their thoughtful assessment and insightful feedback. We are encouraged that they view the finding that FIFO can outperform Reservoir over long training horizons as a noteworthy and counter-intuitive phenomenon in OCSSL. We appreciate their recognition that our experimental evaluation is thorough and systematically explores diverse settings to validate the effectiveness of SOLAR. We address their comments below and have incorporated all feedback into the revised version.
>
> **Weaknesses**
>
> ---
>
> **1)** *Problem Reframing.*
>
> We argue that Latent Rehearsal Decay significantly differs from buffer overfitting as it is known in supervised CL. This is because buffer overfitting only degrades representations and accuracy of samples *outside* of the buffer, while the model achieves good accuracy on buffer samples. We added Figure 14, showing how metric degradation with Reservoir happens similarly for both buffer and all training samples, implying a global feature degradation. In particular, the rise of Overlap for buffer samples in Reservoir makes even buffer samples harder to discern among them. We hypothesize that overfitting on the buffer is a cause for Latent Rehearsal Decay, but does not fully describe the phenomenon. We have added a discussion of the difference between these phenomena in Appendix B.4. Moreover, in Appendix D.2 we show hints of a phenomenon attributable to Latent Rehearsal Decay happening for CLA-R, which we observed having excessive stability, but without overfitting as it employs a FIFO buffer.
>
> ---
>
> **2)** *Incremental Contribution.*
>
> While we agree that SOLAR’s Deviation-aware buffer has the same conceptual essence as PER in prioritizing “hard samples”,  we argue that SOLAR has the following advantages:
> - It has solid insights as to why the hard samples are needed specifically for the OCSSL scenario, namely the analysis on novel metrics and Latent Rehearsal Decay.
>  - How we prioritize hard samples is a crucial challenge: the Deviation-aware buffer focuses on both the buffer update and extraction policies, while other methods, such as PER, focus only on the extraction policy, as the buffer composition is a simple FIFO for PER.
> We have added in Appendix C.6 a comparison of well known methods that prioritize “hard” samples (including PER).
> We have added PER to the experiments--substituting, as suggested, the TD-error with the SSL loss--and, as advised by another reviewer, also LARS [1].
> Full results on these two methods are reported in Table 1 and below in a more compact form.
>
> | Method    | ImageNet (Final / Avg) | CLEAR (Final / Avg) |
> |-----------|------------------------|-------------------|
> | Reservoir | 40.5 / 39.3            | 47.1 / 35.3       |
> | FIFO      | 48.7 / 38.9            | 45.3 / 34.9       |
> | LARS      | 43.2 / 39.2            | 44.7 / 34.1       |
> | PER       | 48.8 / 38.8            | 46.0 / 35.0       |
> | SOLAR     | **49.4 / 42.8**            | **51.5 / 41.3**       |
>
> SOLAR is able to beat both newly added methods. PER performs closely to FIFO, as both share the same buffer update policy, confirming that an extraction policy prioritizing ''hard'' samples is not enough for OCSSL. LARS instead performs similar to Reservoir, as both have insertion probabilities decreasing with time and converge to a fixed subset.
>
> [1] Buzzega, Pietro, et al. "Rethinking experience replay: a bag of tricks for continual learning." 2020 25th International Conference on Pattern Recognition (ICPR). IEEE, 2021.
>
> ---
>
> **3)** *Lack of Deeper Intuition on the Proposed Metrics.*
>
> SVD and uniformity measure a global form of collapse, which fails to capture Latent Rehearsal Decay. Instead, our metrics measure local behavior by computing instance-level measures, which capture a more fine-grained view of representation quality and convergence. We do this by considering the hyperballs defined by feature space views, which give a local insight even for an inter-sample metric such as Overlap. SVD and uniformity fail to address this, while the first only considers global feature collapse, the other considers only the global uniformity of samples in the latent space. We have added a paragraph with these motivations in Section 4.

---

> ### Author Response · Authors · 2025-11-20
> **Response 2/2**
>
> **4)** *Compatibility with other Self-Supervised Algorithms.*
>
> We have added additional experiments using SimCLR instead of SimSiam as the backbone SSL model in Appendix D.5. We report results from newly added Table 7 below.
> | Method      | CIFAR-100 (Final/Avg) | ImageNet100 (Final/Avg) |
> |-------------|-----------------------|-------------------------|
> | ER (Reservoir) | 45.16 / 42.03       | 47.90 / 43.02           |
> | ER (FIFO)      | 46.35 / 41.81       | **49.38** / 42.12       |
> | MinRed         | 43.92 / 43.05       | 45.54 / 43.30           |
> | SCALE          | 32.15 / 27.28       | 36.45 / 31.48           |
> | **SOLAR**      | **46.62 / 43.35**   | 49.10 / **43.53**       |
>
> We observe similar results to SimSiam; FIFO scores higher Final Accuracy and Lower Average Accuracy when compared to Reservoir. SOLAR performance appears intact in the presence of a contrastive loss.
> We also added Figure 19, which reports CIFAR100 probing accuracy of methods across training: SOLAR accuracy curve consistently surpasses that of FIFO and Reservoir across all training. However, the contrastive loss seems to dampen the effect of Latent Rehearsal Decay for Reservoir. Instead of a sudden decrease in accuracy, we see a gentle inflection of accuracy for later stages of training. We hypothesize that this is because the contrastive loss plays a similar role as the Overlap Loss in SOLAR in preventing excessive latent collapse.
>
> ---
>
> **5)** *Lack of Quantitative Computational Analysis.*
>
> We have added computational times of various methods on a single ImageNet100 task and their analysis in Appendix D.4.
> SOLAR has a comparable computational cost to most of the other methods.
>
> ---
>
> **6)** *Incomplete Experimental Analysis.*
>
> Figure 11 reports the SVD for features extracted on the entire test set at the end of training. We have added an analysis of SVD calculated at the beginning (after experience 0), middle (after experience 9) and end (after experience 19) of training in Figure 13. For each, we report separately the SVD lines for past, current and future tasks. We have expanded our analysis in Section B.3 to encompass these additional plots. However, once again plots of FIFO and Reservoir are almost identical, even though the latter suffers from Latent Rehearsal Decay.
>
> ---
>
> **7)** *Motivations for OCSSL.*
>
> While it is true that large-scale pretraining is the ideal setting in which to learn general representations, OCSSL is a general framework that encompasses many popular learning paradigms:
> - online CL, removing the unlikely availability of labels in a streaming setting
> - test-time adaptation, where models are adapted online on unlabeled data
> - many post-training methods are "almost" online since they do few epochs with small minibatch sizes, and they need to prevent forgetting of pretrained knowledge.
>
> Moreover, we envision that OCSSL could be used in settings where data curation is not convenient. Overall, while OCSSL is a restrictive setting, we believe advances in this area are general and they can provide benefits to many real-world applications. We have better explained the importance of OCSSL in Section 1, and, to strengthen our case, we have conducted additional experiments on the iNaturalist dataset, simulating a real world scenario in which a pretrained network is fine-tuned instead of training from scratch (see Section 6.3 and table below).
>
> | Method      | Final Acc. | Avg. Acc. |
> |-------------|------------|-----------|
> | pretrained  | 36.42      | –         |
> | Reservoir   | 42.67      | 40.45     |
> | FIFO        | 43.91      | 40.77     |
> | **SOLAR**   | **44.11**  | **41.73** |
>
> Results show that online Continual pretraining of a network is indeed beneficial as it leads to non-trivial improvements, and that OCSSL techniques are useful in such scenarios.

---

> > ### Comment · Reviewer_fBLe · 2025-11-26
> >
> > Thanks for the authors’ rebuttal. I think the authors have fully addressed my concerns on points 3, 5, and 6; have partially addressed points 1 and 2; but the response on point 7 remains unconvincing to me.
> >
> > Specifically, I find it unclear why the rise of “Overlap” among buffer samples in the Reservoir should make even the buffered samples harder to differentiate. If buffer samples become difficult to distinguish, this seems more like an optimization issue—possibly due to the small batch size or inherent challenges of online learning—rather than a fundamental property of the buffer itself.
> >
> > Regarding point 7, the discussion of an “online pre-trained paradigm” trained entirely from scratch also feels somewhat odd, since in practice it is well known that contrastive-based training typically requires far more iterations than supervised training. Although I agree that the proposed Latent Rehearsal Decay may not simply reflect overfitting, its practical significance remains unclear. Moreover, the paper’s analysis of the underlying causes of Latent Rehearsal Decay still seems vague: it is not fully explained whether the observed overlap arises from insufficient training under online conditions, from the inherently small batch sizes typical in losses like SimCLR, or from issues caused by the buffer mechanism itself. In fact, even the performance gap between Reservoir and SOLAR in Table 7 is rather small, as is the difference between FIFO and SOLAR.
> >
> > While I appreciate the authors’ effort and acknowledge the other reviewers’ positive assessments, I regret to say that the paper still leaves too many open questions to be accepted in its current form. I believe that resolving these issues could make it an insightful and valuable work in the future. Although this paper may already be stronger than several prior OCSSL submissions, that alone should not meet the acceptance standard for ICLR. At present, I do not consider it ready for acceptance.

---

> > > ### Author Response · Authors · 2025-12-03
> > > **Response 1/2**
> > >
> > > We thank the reviewer for the careful response. To facilitate a clearer discussion, we will address inline  each of the points still identified as weak by the reviewer.
> > >
> > > > Thanks for the authors’ rebuttal. I think the authors have fully addressed my concerns on points 3, 5, and 6; have partially addressed points 1 and 2; but the response on point 7 remains unconvincing to me.
> > >
> > > The reviewer says that we partly addressed point 2, but fails to give a coherent motivation for why the response was not satisfactory enough. On our part, we believe that we have addressed Weakness 2 completely, both from theoretical and empirical standpoints.
> > >
> > > > Specifically, I find it unclear why the rise of “Overlap” among buffer samples in the Reservoir should make even the buffered samples harder to differentiate. If buffer samples become difficult to distinguish, this seems more like an optimization issue—possibly due to the small batch size or inherent challenges of online learning—rather than a fundamental property of the buffer itself.
> > >
> > > The reviewer finds the relation between Overlap and sample differentiation unclear. We argue that Overlap is *precisely* a measure of sample differentiation: in Section 4 we define it as a metric that quantifies how distinguishable two samples are by measuring the separation between their feature hyperballs.
> > > Moreover, we affirm that rising Overlap in Latent Rehearsal Decay is **not** linked to the online training setting, as it is amply demonstrated in our paper that the phenomenon does not happen when using other buffers, e.g. FIFO. The entirety of Sections 3 and 4 of our paper are devoted to explaining this and highlighting the excessive stability of Reservoir as a cause for Latent Rehearsal Decay. Exploring if–under certain conditions– Latent Rehearsal Decay appears even in offline scenarios is out of the scope of this paper and offline training dynamics significantly change how this problem could be averted.
> > >
> > > > Moreover, the paper’s analysis of the underlying causes of Latent Rehearsal Decay still seems vague:  it is not fully explained whether the observed overlap arises from insufficient training under online conditions, from the inherently small batch sizes typical in losses like SimCLR, or from issues caused by the buffer mechanism itself. In fact, even the performance gap between Reservoir and SOLAR in Table 7 is rather small, as is the difference between FIFO and SOLAR.
> > >
> > > We provided an extended analysis of Latent Rehearsal Decay and its causes in Sections 4 and Appx. B, clearly stating that it is caused by the excessive stability of buffer dynamics–as in Reservoir buffer.
> > > We argue that short training is **not** a cause for Latent Rehearsal Decay, on the contrary, the phenomenon arises for longer training. This is clearly visible in Figures 2, 3 and 6, and discussed in Sections 3, 4 and 6.
> > > Regarding contrastive losses (e.g. SimCLR loss), we had presented additional experiments in the revision –as requested in the reviewer’s initial review. We discuss in Appendix that the phenomenon of Latent Rehearsal Decay appears dampened when using SimCLR loss, this implies that the improvement that our method can provide (reported in Table 7) are consequently reduced. SimCLR is suboptimal for OCSSL as its performance is largely dependent on the minibatch size [1], for this reason the simpler method SimSiam is commonly employed in OCSSL literature [2,3,4] and we followed this paradigm in our main experiments, in order to disentangle the uncovered phenomena from batch size.

---

> > > ### Author Response · Authors · 2025-12-03
> > > **Response 2/2**
> > >
> > > > Regarding point 7, the discussion of an “online pre-trained paradigm” trained entirely from scratch also feels somewhat odd, since in practice it is well known that contrastive-based training typically requires far more iterations than supervised training. Although I agree that the proposed Latent Rehearsal Decay may not simply reflect overfitting, its practical significance remains unclear.
> > >
> > > We fail to understand the reviewer’s motivations behind their criticism of the OCSSL setting. First, the comparison with supervised does not make sense to us, as the absence of labels is a scenario characteristic and not a methodological choice. Secondly, the reviewer wrongly claims that long training is a requirement for SSL: SSL methods continue to improve for hundreds of epochs, but quality results can be obtained with shorter training too [1, 5, 6] –SimSiam itself has this property– and there exists SSL methods tailored to work in short training settings [7].
> > > We have discussed the practical application of OCSSL in both previous responses and in Section 1, and also added in the rebuttal empirical proof that methods and scenarios are applicable in a real-world setting.
> > > We have highlighted the importance of Latent Rehearsal Decay in Sections 1, 3 and 4, in substance its importance derives from the fact that applying common CL techniques such as Reservoir can be unintuitively detrimental in the OCSSL scenario.
> > > Finally the reviewer says that he is now convinced that Latent Rehearsal Decay is different from buffer overfitting, but previously stated that we responded only partially to Weakness 1 without further expanding. This seems incoherent and it makes us incapable of understanding or addressing the eventual criticism.
> > >
> > > > While I appreciate the authors’ effort and acknowledge the other reviewers’ positive assessments, I regret to say that the paper still leaves too many open questions to be accepted in its current form. I believe that resolving these issues could make it an insightful and valuable work in the future. Although this paper may already be stronger than several prior OCSSL submissions, that alone should not meet the acceptance standard for ICLR. At present, I do not consider it ready for acceptance.
> > >
> > > While we thank the reviewer for having grasped some of the strengths of our paper, we believe that the reviewer score reflects a subjective disagreement –not shared by the wider CL research community– on a research area, specifically Online Continual Self-Supervised Learning, and not a problem of our work. Following their reasoning, any research direction that does not show immediate practical benefits should be rejected from ICLR, even when it provides significant advancements compared to the literature. This approach would result in rejecting most CL papers.
> > >
> > > [1] Chen, Ting, et al. "A simple framework for contrastive learning of visual representations." International conference on machine learning. PmLR, 2020.
> > >
> > > [2] Cignoni, Giacomo, et al. "CLA: Latent Alignment for Online Continual Self-Supervised Learning." arXiv preprint arXiv:2507.10434 (2025).
> > >
> > > [3] Cignoni, Giacomo, et al. "Replay-free Online Continual Learning with Self-Supervised MultiPatches." arXiv preprint arXiv:2502.09140 (2025).
> > >
> > > [4] Purushwalkam, Senthil, et al. "The challenges of continuous self-supervised learning." European conference on computer vision. Cham: Springer Nature Switzerland, 2022.
> > >
> > > [5] He, Kaiming, et al. "Momentum contrast for unsupervised visual representation learning." Proceedings of the IEEE/CVF conference on computer vision and pattern recognition. 2020.
> > >
> > > [6] Chen, Xinlei, et al. "Exploring simple siamese representation learning." Proceedings of the IEEE/CVF conference on computer vision and pattern recognition. 2021.
> > >
> > > [7] Tong, Shengbang, et al. "Emp-ssl: Towards self-supervised learning in one training epoch." arXiv preprint arXiv:2304.03977 (2023).

---

### Official Review · Reviewer_FVaz · 2025-11-01

**Soundness:** 3
**Presentation:** 2
**Contribution:** 3
**Rating:** 6
**Confidence:** 4

**Summary:**

This paper investigates the learning dynamics of online continual self-supervised learning (OCSSL) and identifies it to favor plasticity over stability. Through empirical analysis, the authors make the observation that FIFO buffer management outperforms reservoir sampling in OCSSL settings. They attribute this to a phenomenon they term "latent rehearsal decay," which arises from prolonged training on static subsets of data during replay. To address this, the paper introduces two main contributions: (1) two metrics to quantify latent rehearsal decay, and (2) a solution framework consisting of SOLAR, a deviation-aware replay buffer management strategy that prioritizes samples with higher loss, and an overlap loss for regularization. The method is evaluated using Simple Siamese networks as the SSL backbone and demonstrates improved accuracy over long training horizons.

**Strengths:**

- The identification of latent rehearsal decay as a specific challenge in OCSSL is novel, and the counterintuitive finding that FIFO outperforms reservoir sampling in this setting provides fresh perspective on the plasticity-stability trade-off.
- The core technical insight is sound—that samples with higher loss indicate higher feature deviation and should be prioritized for replay. The proposed metrics for quantifying latent rehearsal decay provide a useful analytical tool for the community.
- The provided solutions show practical improvements in accuracy over long training horizons. The deviation-aware buffer management strategy is a valuable contribution that could be adapted to various continual learning scenarios.

**Weaknesses:**

- The evidence that plasticity is favored rather than stability rests primarily on FIFO vs. reservoir comparison, which may not be sufficiently comprehensive, given that this comparison fails in the case of CLEAR dataset. This comparison seems to work for Class incremental online ssl but not in domain incremental online ssl.

- Figure 2(a) has a confusing x-axis. The explanation of what is meant by mini batch passes is unclear as a measure of training schedule. The plot shows in Imagenet100  reservoir achieves accuracy of >46 after one minibatch pass but this is contradictory with Figure 3 where it acheives >46 accuracy after several passes

- Figure 2(b) does not clarify which dataset the results are based on.

- Motivation for why OCSSL is an important problem area could be better articulated.

**Questions:**

-  Why is EMA necessary for updating buffer statistics rather than direct replacement using the current values ?
- What is the specific contribution of extraction counts, why can't we just select based on loss ?
- Can you clarify the apparent inconsistency between Figure 2(a) showing >46% accuracy after 1 mini-batch pass and Figure 3 showing this accuracy is reached only after 50k steps on ImageNet?

---

> ### Author Response · Authors · 2025-11-20
> **Response 1/2**
>
> We thank the reviewer for their careful evaluation. We are encouraged that they see Latent Rehearsal Decay as offering a fresh perspective on replay in OCSSL and that our findings contributes to understanding plasticity–stability dynamics. We are pleased that they find the Deviation and Overlap metrics valuable and that they recoginze SOLAR practical improvements over long training horizons across Continual Learning setups. We address their comments below and have incorporated all feedback into the revised paper.
>
> **Weakenesses**
>
> ---
>
> **1)** *Evidence that plasticity is favored rather than stability.*
>
> We added clarifications in Section 1 and 4 stating that the missing plasticity of Reservoir induces Latent Rehearsal Decay only in scenarios where plasticity is needed. The goal of our paper is *not* to generalize the lack of plasticity or Latent Rehearsal Decay to all OCSSL scenarios, but instead Latent Rehearsal Decay is analyzed as a failure case of typical methods when applied in *some* OCSSL setting. In fact, the phenomenon does not appear on short training runs even if class-incremental. Lack of plasticity and Latent Rehearsal Decay are strong motivation for why a novel method (namely SOLAR) capable of dynamically adapting to various requirements of stability and plasticity is needed.
>
> The CLEAR dataset, similar to online class-incremental learning with shorter training runs, is one of those scenarios where stability is preferred; this is demonstrated by Reservoir performing well in this scenario. We hypothesize that a domain incremental scenario (especially one with a comparatively weaker domain shift such as CLEAR) favors stability because successive tasks contain similar data and thus the model does not need to adapt as much to newer data.
> Nonetheless, SOLAR is able to score well on this scenario thanks to its adaptive plasticity capabilities.
> We added this clarification in Section 6.1.
>
> ---
>
> **2)** *Figure 2(a) has a confusing x-axis.*
>
> We apologize for the misunderstanding. By minibatch passes we intend to have multiple training steps (each with different replay samples from the buffer) for each incoming minibatch from the stream. In plots in Figure 2(a) we report results for 1-6 minibatch passes, where 1 minibatch pass corresponds to a model trained with fewer training steps (and thus short training) while 6 minibatch passes corresponds to more overall training steps (longer training). We have added a better explanation of this in the caption of Figure 2(a).
>
> ---
>
> **3)** *Figure 2(b) does not clarify which dataset the results are based on.*
>
> We have clarified in the caption that results for Figure 2(b) are calculated on CIFAR100.
>
> ---
>
> **4)** *Motivation for why OCSSL is an important problem area.*
>
> We have better explained the importance of OCSSL in Section 1, stating that OCSSL is needed in the context of online CL, as it is unrealistic to envision a scenario with online streaming data but also labels, especially human provided ones. Moreover, we have added experiments in a real-world scenario, using a pretrained network to continually train on iNaturalist (see Section 6.3, results also reported below).
> | Method      | Final Acc. | Avg. Acc. |
> |-------------|------------|-----------|
> | pretrained  | 36.42      | –         |
> | Reservoir   | 42.67      | 40.45     |
> | FIFO        | 43.91      | 40.77     |
> | **SOLAR**   | **44.11**  | **41.73** |
>
> This new experiment demonstrates that OCSSL can improve pretrained networks on real-world data. Equipped with SOLAR, OCSSL methods can thus provide a streaming training strategy with no requirements on labels.

---

> ### Author Response · Authors · 2025-11-20
> **Response 2/2**
>
> **Questions**
>
> ---
>
> **1)** *Why is EMA necessary for updating buffer statistics rather than direct replacement using the current values?*
>
> The use of EMA-updated metrics is already incorporated in Continual SSL methods; [1] and [2] employ EMA to track sample features stored in the buffer. Intuitively, the EMA of statistics allows estimating them online during training and is robust to sudden distribution shifts. We added this explanation to section 5.1 and ablation of the contribution of EMA-updated buffer statistics in Table 4 (Appendix C.5, we also report results below).
>
> | Method | EMA 𝓜 Stats | CIFAR-100 (Final/Avg.) | ImageNet100 (Final/Avg.) |
> |--------|-------------|-------------------------|---------------------------|
> | SOLAR  | ✗           | 49.0 / 41.3             | **49.4** / 42.0           |
> | **SOLAR** | **✓**    | **49.5 / 42.3**         | **49.4 / 42.8**           |
>
> We observe a small but tangible improvement for EMA-updated statistics over directly replacing the statistics with their updated version. While not a core component of our approach, it nonetheless strengthens the robustness of SOLAR.
>
> [1] Purushwalkam, Senthil, et al. "The challenges of continuous self-supervised learning." European conference on computer vision. Cham: Springer Nature Switzerland, 2022.
>
> [2] Cignoni, Giacomo, et al. "CLA: Latent Alignment for Online Continual Self-Supervised Learning." arXiv preprint arXiv:2507.10434 (2025).
>
> ---
>
> **2)** *What is the specific contribution of extraction counts, why can't we just select based on loss?*
>
> We consider buffer sampling based on extraction count as a complementary method of training on high Deviation samples, while having a balancing effect that prevents overfocusing on a small set of samples. Intuitively, samples that have been replayed fewer times have higher Deviation and having a different approximation for Deviation than loss makes the method more robust. We have included clearer motivations for the use of replay selection based on buffer sampling in Section 5.1, including these considerations. Also, we have conducted an ablation in Appendix C.5 comparing random, loss, and extraction count policies, which also we report here.
>
> | Extraction Method                  | CIFAR-100 (Final/Avg.) | ImageNet100 (Final/Avg.) |
> |-----------------------------------|--------------------------|----------------------------|
> | SOLAR + random extraction         | 48.7 / 42.2              | 43.3 / 42.0                |
> | SOLAR + $ℓᵢ^𝓜$ extraction           | 49.4 / 42.0              | 41.9 / 42.3                |
> | **SOLAR + $eᵢ^𝓜$ extraction**       | **49.5 / 42.3**          | **49.4 / 42.8**            |
>
>
> On CIFAR-100, the extraction count selection for replay gives only a marginal improvement, instead it significantly surpasses other policies on ImageNet100.
>
> ---
>
> **3)** *Can you clarify the apparent inconsistency between Figure 2(a) showing >46% accuracy after 1 mini-batch pass and Figure 3 showing this accuracy is reached only after 50k steps on ImageNet?*
>
> We believe that we have addressed this concern in our response to weakness 2.

---

> > ### Comment · Reviewer_FVaz · 2025-11-26
> >
> > I thank the authors for their rebuttal and the revisions made. The updates have strengthened the paper’s presentation, and I am therefore increasing the presentation score while keeping the remaining scores as originally assigned.

---

> > > ### Author Response · Authors · 2025-12-03
> > >
> > > We thank the reviewer for their insightful feedback which helped us improve the clarity and strength of our paper.

---

### Official Review · Reviewer_HQxo · 2025-11-01

**Soundness:** 3
**Presentation:** 2
**Contribution:** 2
**Rating:** 6
**Confidence:** 4

**Summary:**

The paper studies identifies a novel challenge in  Online Continual Self-Supervised Learning (OCSSL), Latent Rehearsal Decay, where replay strategies that converge to a static subset (*e.g.*, Reservoir) degrade latent representations over long online training streams, degrading downstream accuracy. To quantify this phenomenon, the authors introduce two latent-space metrics, Deviation (intra-sample variability in augmented features) and Overlap (inter-sample feature intersection), and show that these metrics correlate with drops in accuracy. Building on this analysis, they propose SOLAR (Self-supervised Online Latent-Aware Replay), which *i)* uses a Deviation-aware buffer that prioritizes high-loss samples and *ii)* an Overlap loss that penalizes positive overlap between the features from the current minibatch and the ones from the buffer. Experiments on Split CIFAR-100, ImageNet-100, and CLEAR100 show that SOLAR achieves strong average and final accuracy after linear probing, mitigating the reported Latent Rehearsal Decay.

**Strengths:**

- Latent Rehearsal Decay represents a meaningful and interesting challenge in Online Continual Self-Supervised Learning.

- The proposed method demonstrates competitive performance across diverse benchmarks (CIFAR-100, ImageNet-100, CLEAR100), buffer configurations, and training lengths.

- The paper is well-written and easy to follow in its intuition and formulation.

**Weaknesses:**

1. The Deviation-aware buffer closely resembles prior loss- and gradient-based storage and retrieval strategies. The paper omits important references such as Gradient-based Sample Selection (GSS) (Aljundi et al., NeurIPS 2019, gradient-aware storage), Rethinking Experience Replay (LARS) (Buzzega et al., ICPR 2020, loss-aware storage), and Maximally Interfered Retrieval (MIR) (Aljundi et al., NeurIPS 2019, interference-aware retrieval). These works define related mechanisms for sample storage and retrieval; a theoretical and empirical comparison would clarify the novelty and advantages of the proposed approach.

2. Figures are poorly integrated into the text, as they are referenced only after extended discussion, and multi-panel figures have their subcomponents explained in separate sections. Captions are often too long and mix description with interpretation, which fragments the narrative and reduces readability.

3. The computation of Deviation and Overlap requires generating multiple augmentations per sample. The paper does not quantify the online computational cost or analyze the sensitivity of SOLAR to its hyperparameters. A complexity and sensitivity analysis would improve the understanding of SOLAR’s applicability robustness.

4. Reference works for continual learning, online continual learning, and self-supervised learning are mis-cited: when first mentioning these subjects, the manuscript relies primarily on recent secondary papers instead of referring to foundational works.

**Questions:**

1. Are the Deviation and Overlap curves in Figure 3 computed from the training stream, from the buffer only, or current minibatch + buffer?

2. When the Overlap loss is applied on top of Reservoir or FIFO, the paper mentions inconsistent (and sometimes negative) gains. What's your intuition on the reasons behind this?

---

> ### Author Response · Authors · 2025-11-20
> **Response 1/2**
>
> We thank the reviewer for their constructive feedback. We are encouraged that they view Latent Rehearsal Decay as a meaningful phenomenon in OCSSL, and appreciate their positive assessment of our motivations, formulations, and overall soundness. We are also pleased that they recognize the competitiveness and robustness of SOLAR across experimental settings as well as the value of our empirical study and latent-space metrics. We address their comments below and have integrated all feedback into the revised paper.
>
> **Weaknesses**
>
> ---
>
> **1)** *The Deviation-aware buffer resembles prior strategies*
>
> Although the idea of prioritizing “hard” samples already exists in the supervised CL literature, our proposal differs both on theoretical and practical grounds and is tailored to needs of the OCSSL scenario. We added a thorough comparison of SOLAR with MIR, GSS, LARS, and, as suggested by another reviewer, PER [1] in Appendix C.6.
>
> Essentially, how hard samples are prioritized varies significantly from method to method.  MIR uses a plain Reservoir buffer, but extracts samples with maximal loss interference. GSS instead stores samples with maximal gradient diversity but with a random extraction policy. LARS instead extends Reservoir by including loss-based deletion in case an insertion is needed, but its insertion policy is the same as Reservoir, meaning that the insertion probability decreases with time and converges to a fixed subset.
>
> Some of these methods are not practical for the OCSSL scenario: MIR requires a forward pass of the entire buffer at each step, which is excessively costly for OCSSL while GSS requires additional costly backward pass of buffer samples to calculate gradients. LARS, instead, is applicable to OCSSL without excessive computational burdens.
>
> Instead, the Deviation-Aware component of our proposed method SOLAR is based on theoretical grounds and is tailored to the needs of OCSSL. The Deviation-Aware buffer, differing from other methods, jointly addresses both buffer update and extraction policies by keeping only the highest loss sample available at each moment and replaying based on the lowest soft extraction count. Thus, it incurs no costly computation overhead and is capable of both focusing on “hard” samples and encouraging diversity in a way that is coherent with the underlying needs of OCSSL and motivated by our analysis on the novel metrics and Latent Rehearsal Decay.
>
> We also repeated all main experiments on LARS and PER and added them to Table 1. We report below a significant portion of Table 1 for convenience.
>
> | Method    | ImageNet (Final / Avg) | CLEAR (Final / Avg) |
> |-----------|------------------------|-------------------|
> | Reservoir | 40.5 / 39.3            | 47.1 / 35.3       |
> | FIFO      | 48.7 / 38.9            | 45.3 / 34.9       |
> | LARS      | 43.2 / 39.2            | 44.7 / 34.1       |
> | PER       | 48.8 / 38.8            | 46.0 / 35.0       |
> | SOLAR     | **49.4 / 42.8**            | **51.5 / 41.3**       |
>
> SOLAR performs better than the new methods. Moreover, LARS performs similarly to Reservoir, as its update dynamic closely matches those of Reservoir. PER instead closely matches the performances of FIFO, as they uses the same policy for buffer update.
>
> [1] Schaul, Tom, et al. "Prioritized experience replay." International Conference on Learning Representations (2016).
>
> ---
>
> **2)** *Figures poorly integrated into the text*
>
> We have moved Figure 2 at the bottom of page 3 and Figure 3 at the top of page 5 in order to improve readability. We also improved the caption of Figure 2 and 3.
>
> ---
>
> **3)** *Metrics computations, hyperparameters sensitivity and complexity*
>
> The analysis of Overlap and Deviation in Figure 3 is done offline on model checkpoints as an evaluation measure. The online estimate of Deviation and Overlap in SOLAR comes at no additional cost as we employ the features of the two augmented views already extracted by the SSL method, as explained in Section 5.2. Comparison between online Overlap estimation and true offline Overlap is presented in Appendix C.3. Hyperparameter ablations and analysis are given in Appendix C.4. In response to the reviewer comment, we now additionally include a training time analysis in Appendix D.4 showing that SOLAR has similar time complexity to other methods.
>
> ---
>
> **4)** *Reference works for continual learning*
>
> We added additional citations for foundational works in CL, online CL and SSL in Section 1.

---

> ### Author Response · Authors · 2025-11-20
> **Response 2/2**
>
> **Questions**
>
> ---
>
> **1)** *Are the Deviation and Overlap curves in Figure 3 computed from the training stream, from the buffer only, or current minibatch + buffer?*
>
> Curves from Figure 3 show metrics computed on the entire training stream, we added a clarification of this in the caption and in Section 4.
>
> ---
>
> **2)** *When the Overlap loss is applied on top of Reservoir or FIFO, the paper mentions inconsistent (and sometimes negative) gains. What's your intuition on the reasons behind this?*
>
> Our intuition is that these buffer strategies create conditions under which the Overlap loss cannot operate as intended. In the case of Reservoir, the buffer already contains a small fixed set of examples that are well learned and also have low Deviation. Iteratively enforcing low Overlap over an already learned fixed set of samples could be redundant and even amplify the negative effects of Reservoir. For FIFO, the buffer is dominated by the most recent samples, biased towards the latest task. As a result, the Overlap loss mainly encourages separation among examples coming from similar distributions while failing to regulate overlap across different tasks. We have added a paragraph expanding on this hypothesis in section 6.2.

---

> > ### Comment · Reviewer_HQxo · 2025-11-26
> >
> > I thank the authors for their response and for their effort. I have reviewed everything carefully and decided to keep my original overall score, which remains positive. I have nonetheless increased both my contribution score and my confidence score, as the authors’ clarifications helped me better appreciate several aspects of the work.
> >
> > My reasoning is that, considering the other reviews as well, there is still room to further refine the positioning of the work with respect to the existing literature. This is purely a matter of presentation and framing: the paper is good, offering many insights and a solid set of experiments with valuable contributions. However, it should be clearer in the main paper why the proposed problem departs from the existing literature, whether the proposed OCSSL setting constitutes a truly impactful scenario, and what the precise technical contributions are (especially since, as the authors themselves note, some ideas are borrowed from prior CL literature and the deviations, while present, appear somewhat subtle). All these points were addressed well in the authors’ responses, but I am not fully convinced they are equally evident in the updated submission.

---

> ### Author Response · Authors · 2025-12-03
>
> We thank the reviewer for their constructive remarks that enabled us to refine and improve our work.
> We regret the recent circumstances at ICLR that interrupted an otherwise fruitful discussion. Additional details regarding the presentation issues the reviewer identified would have allowed us to further refine the clarity of the paper presentation and framing.

---

### Author Response · Authors · 2025-12-03
**Summary of interactions for Area Chair**

We sincerely thank the Area Chairs and all reviewers for their dedicated effort. We write to briefly summarize our exchanges so far. The constructive feedback from the reviewers, combined with the significant experimental and expository enhancements we have incorporated, highlight the value and robustness of our contribution.

### **Reviewer HQxo (Rating 6):** Clarifications led to improvement in contribution
The reviewer asked (1) to extend the relation of SOLAR to existing methods that prioritize “hard” buffer samples, (2) improve the presentation of some figures and missing important citations, (3) better analysis of the ablation of Overlap loss paired with other buffer and (4) sensitivity and complexity analysis.
Following their observations, (1) we added an extended discussion comparing SOLAR –which is theoretically grounded for OCSSL– with other conceptually similar methods and added two of those to experimental comparisons, showing that they perform worse than SOLAR, (2) improved citations and figures presentation, (3) expanded the discussion on Overlap loss paired with Reservoir and FIFO, (4) highlighted already present sensitivity analysis and added computational times comparison.
Overall the reviewer seems satisfied with our revisions, saying that he improved the contribution score.

### **Reviewer FVaz (Rating 6):** Satisfied with responses which improved paper presentation
The reviewer (1) asked for clarification on the contributions of EMA-updated metrics and buffer extraction strategy, (2) discussion on the plasticity-stability tradeoff for OCSSL, (3) asked for more motivation on OCSSL importance and (4) for improving clarity of some figures.
During rebuttal we strengthened our paper with (1) requested ablations and discussions on mentioned components, (2) discussion regarding stability and plasticity requirements for OCSSL, (3) expanded on OCSSL importance and provided a real-world pretraining scenario of OCSSL with iNaturalist and (4) improved clarity of mentioned figures.
We are grateful to the reviewer for having pointed out potential unclear points in our paper; overall, the reviewer seemed satisfied with our revision and improved our presentation score.

### **Reviewer fBLe (Rating 4):** Revision strengthens papers but unconvinced on the OCSSL scenario
The reviewer (1) argues that Latent Rehearsal Decay could be associated with buffer overfitting, (2) similar to reviewer HQxo, argues that SOLAR buffer strategy is similar to existing strategies prioritizing “hard samples”, asks for (3) more intuitions on the proposed metrics, (4) compatibility with contrastive SSL losses, (5) computational times, (6) deeper analysis on features SVD, (7) doubts about the usefulness of the OCSSL scenario.
We addressed (2) and (5) as explained with previous reviewers and we added discussion and experiments for both features SVD (6) and intuitive explanations for (3), which fully satisfied the reviewer. Regarding point (1), we provided proof that latent space degradation happens also for the buffer in Latent Rehearsal Decay, the reviewer says in different parts of their response both that “he is convinced” and that “it partly addresses concerns” regarding our revision, without further explanations. We addressed point (7) with extensive motivations, both in the revised paper and Openreview response, and with real-world experiments with pretraining on iNaturalist, but the reviewer still disagrees conceptually on the scenario.
Moreover, in the response to our rebuttal, the reviewer raised new questions, specifically regarding short training or online scenario as causes for Latent Rehearsal Decay, performance with contrastive losses and the role of Overlap. Contrary to the initial review –that analyzed in depth our paper– these questions concern aspects which were in our opinion already very clear in the paper (such as Latent Rehearsal Decay appearing in long training, and Overlap measuring of sample differentiation), or aimed at criticizing the online scenario itself. For these reasons, we did not modify the paper after this second round of questions and only responded on OpenReview.
Although we thank the reviewer for the thoughtful and insightful initial review, we find the second response to be biased and hasty, even raising questions that were clear in the initial version of the paper.

---

### Meta-Review · Area_Chair_D3cY · 2026-01-07

**Summary:**

The paper studies Online Continual Self-Supervised Learning (OCSSL) and identifies a long-horizon failure mode termed Latent Rehearsal Decay.

The reviewers find the phenomenon interesting and the empirical study fairly extensive.
Two reviewers (HQxo, FVaz) remain positive on the borderline margin with score 6.
They highlighted that the problem is meaningful and the method performs competitively in the report results. While maintaining the original score 6, they still hold the doubt on the clear novelty of the work (whether the work significantly differs from existing work), the real impact of the work, and precious technical contribution. Reviewer fBLe is more critical and questions the novelty and practical significance, arguing that the phenomenon may overlap with other technical aspects, such as known buffer overfitting ideas, and that the motivation for OCSSL as an impactful setting is not fully convincing.

The AC is also convinced that, while the topic and observations are interesting, there remain substantial open questions and unclear aspects that prevent acceptance at this stage. In particular, the relationships between the observed phenomenon, the proposed metrics, and the individual technical components are not sufficiently disentangled, making it difficult to clearly attribute the reported gains and to assess the validity of the conclusions. The AC therefore recommends rejection and encourages the authors to carefully revise and strengthen the work.

**Reviewer Concerns:**

The main concerns center on the unclear scope and framing of the paper as a whole. While the rebuttal addresses several specific technical points, such as computational overhead (HQxo, fBLe), hyperparameter sensitivity (HQxo), and presentation and motivation issues raised by FVaz, these improvements do not fully resolve the broader concerns.

Several substantive issues remain outstanding across multiple reviews. These include the clarity of the paper’s positioning and framing (HQxo), the strength of evidence supporting the claim that plasticity is favored, particularly the reliance on FIFO versus Reservoir comparisons (FVaz), and, most critically, Reviewer fBLe’s concerns regarding the core novelty, the ambiguous roles and interactions of the proposed technical components, and the motivation and practical significance of the OCSSL setting itself.

**Reviewer Scores:**

- HQxo: Maintain the original 6. While several technical concerns were addressed in the rebuttal, key issues regarding framing and positioning remain, making further score improvement unlikely.


- FVaz: Likely to maintain the original score 6. The remaining concerns primarily relate to the scope of the claims and the strength of evidence across different scenarios. It is unlikely to justify an increase from 6 to 7 without additional core experiments beyond those already provided.


- fBLe: Maintain the original score of 4. The reviewer explicitly stated that the paper is not ready for acceptance, highlighting unclear practical significance and too many unresolved open questions.

---

### Decision · Program_Chairs · 2026-01-26

Reject